# Effect of Mixed *Lactiplantibacillus plantarum*- and *Bacillus subtilis*-Fermented Feed on Growth, Immunity, and Intestinal Health of Weaner Pigs

**Jun Chen [1], Guofang Wu [2], Huili Pang [3], Jiyun Hua [4], Yifei Guan [3], Miao Zhang [3], Yaoke Duan [3], Guangyong Qin [1], Lei Wang [2,\*], Yimin Cai [5] and Zhongfang Tan [3,\*]**

[1] Henan Key Laboratory of Ion-Beam Bioengineering, School of Physics, Zhengzhou University, Zhengzhou 450052, China; chenjun6377@163.com (J.C.); qinguangyong@zzu.edu.cn (G.Q.)

[2] Plateau Livestock Genetic Resources Protection and Innovative Utilization Key Laboratory of Qinghai Province, Key Laboratory of Animal Genetics and Breeding on Tibetan Plateau, Ministry of Agriculture and Rural Affairs, Qinghai Academy of Animal and Veterinary Medicine, Qinghai University, Xining 810016, China; jim963252@163.com

[3] School of Agricultural Sciences, Zhengzhou University, Zhengzhou 450052, China; pang@zzu.edu.cn (H.P.); guanyifei0904@163.com (Y.G.); miaozhang@zzu.edu.cn (M.Z.); duyk@nwafu.edu.cn (Y.D.)

[4] Haidong City Animal Disease Prevention and Control Center, Haidong 810600, China; huajiyun123@163.com

[5] Japan International Research Center for Agricultural Sciences, Crop, Livestock and Environment Division, Tsukuba 305-8686, Japan; cai@affrc.go.jp

\* Correspondence: wanglei382369@163.com (L.W.); tzhongfang@zzu.edu.cn (Z.T.)

**Abstract:** Bamei pigs, an eximious local breed reared on the Tibetan Plateau of China, are facing problems such as feed shortages, weaning stress, and antibiotic abuse. This study aimed to improve the quality of feed, growth performance, intestinal microbiota, and immunity of Bamei pigs through feeding with potentially probiotic-fermented feed. Different feeds were administered to weaned Bamei piglets for 60 days, creating the following five experimental treatment groups: basal feed group; *Lactiplantibacillus plantarum*-fermented-feed group; *Bacillus subtilis*-fermented-feed group; mixed-fermented-feed group; and antibiotic-supplemented-feed group. The results showed that the pH, neutral detergent fiber, and acid detergent fiber of the potentially probiotic-fermented feeds were significantly reduced; organic acids were produced; and Coliform bacteria, *Clostridium*, and aerobic bacteria were effectively inhibited. Feeding with potentially probiotic-fermented feed not only reduced the feed-conversion ratio but also improved immunity by increasing the production of anti-inflammatory cytokines, as well as decreasing pro-inflammatory cytokines and typical inflammatory pathways. The 16s rDNA high-throughput-sequencing results showed that probiotic-fermented feed improved the diversity of intestinal microbiota, inhibited the growth of the opportunistic pathogens *Clostridium* and *Streptococcus*, increased the relative abundance of *Lactobacillus* and *Prevotella*, and promoted gut health, demonstrating the promising application prospects of potentially probiotic-fermented feed.

**Keywords:** Bamei pig; *Lactobacillus*; *Bacillus*; fermented feed; intestinal microbiota; cytokines; immunity

## 1. Introduction

In 1946, Moore et al. [1] found that adding the right amount of antibiotics to feed can not only effectively reduce the probability of broiler disease but also promote growth. This discovery quickly increased the application of antibiotics as an additive in feed and led to a rapid rise in the number and types of antibiotics used in animal breeding, especially in enhancing animals' disease resistance and reducing the mortality rate due to clinical diseases, effects which have been fully proved [2]. However, antibiotics have also brought about many problems, such as antibiotic resistance, drug residues, and the degradation of pork quality [3,4]. More seriously, the overuse of antibiotics in farmed animal husbandry has

led to the frequent emergence of drug-resistant strains of human and animal pathogens [5]. As a result, many countries and regions have explicitly restricted the use of antibiotics in livestock farming [6], so the livestock and feed industries have entered an era of the stringent banning, restriction, and replacement of antibiotics. This has inspired researchers worldwide to pursue the development of various antibiotic substitutes and harmless green feed additives in order to eliminate or mitigate the negative effects of antibiotic use.

Probiotics and their metabolites are ideal alternatives to antibiotics and positively affect disease prevention and the performance of livestock and poultry. Probiotics can effectively reduce the resistance and toxicity caused by antibiotics, oppose colonization by foreign pathogens, and contribute to improving feed digestibility [7]. Probiotics promote the health of an organism mainly by competing for nutrients, inhibiting the growth of pathogens, producing bacteriocins and vitamins, participating in the intestinal barrier, reducing inflammation, and regulating innate immunity, among other activities [8,9]. Therefore, the application of probiotics in the feedlot industry has long-term developmental prospects. According to the definition provided by the Joint Expert Group of the Food and Agriculture Organization of the United Nations (FAO) and the World Health Organization (WHO), probiotics are "live microorganisms that, when administered in adequate amounts, confer a health benefit on the host" [10].Common probiotics include *Lactobacillus*, *Bifidobacterium*, and some Gram-positive cocci [11,12]. As one of the main probiotics, lactic acid bacteria (LAB) are often used in animal feed to provide relief from diarrhea and resistance to pathogens in feeder animals. On the one hand, LAB compete with diarrhea-causing pathogens for nutrient and intestinal adhesion sites, which increases the number of beneficial bacteria. On the other hand, LAB can also produce organic acids, bacteriocins, and other antimicrobial substances [13–17], inhibiting the growth of diarrhea-triggering pathogens, alleviating diarrhea, and improving resistance to diseases [18]. The *Bacillus* genus also comprises species considered as probiotics such as some strains of *Bacillus subtilis* and *Bacillus coagulans*, which are commonly applied via direct addition and feed fermentation and can produce a variety of antimicrobials, alleviate diarrhea caused by *Escherichia coli*, etc. The secretion of extracellular enzymes can improve the digestibility of the nutrients in animal feed, regulate the gastrointestinal tract of the animal, improve resistance to disease, promote growth, etc. [19,20]. Probiotics can not only inhibit the growth of pathogens by producing immunologically active substances and competing for nutrients, but also they can regulate the intestinal microbiota and regulate immune-related cytokines, inhibit the expression of pro-inflammatory factors, and participate in the construction of intestinal barriers, thus regulating the host's immune system and enhancing the intestinal defense and immunity of the organism [9,21].

Despite the many previous studies describing the probiotic effects of LAB and *Bacillus* in fermented feeds, the mechanisms of action are not fully understood. The aim of this study was to investigate the effects of fermentation with potential probiotic *Lactiplantibacillus* (L.) *plantarum* QP28-1a and *Bacillus* (B.) *subtilis* QB8a on feed quality, growth performance, intestinal microbiota, and immunity of Bamei pigs in order to assess the benefits of feeds fermented with potential probiotics and the effectiveness of replacing antibiotics. The study also provides a theoretical basis for the green and efficient breeding of Bamei pigs.

## 2. Materials and Methods

### 2.1. Strain and Fermentation of Feed

*L. plantarum* QP28-1a and *B. subtilis* QB8a used for fermenting the feed were provided by the Henan Key Laboratory of Ion-Beam Bioengineering of Zhengzhou University. They were isolated and screened from the feces of Bamei pigs by the research team and improved using mutagenesis, resulting in *L. plantarum* QP28-1a displaying broad-spectrum bacteriostatic activity and excellent tolerance and safety [22], and *B. subtilis* QB8a showing high proteolytic enzyme and cellulase activity. The strains were activated and prepared according to Pang et al. [23] and Jeon et al. [24].

Basal feed: refers to the standard and improved preparation of the NY/T 65-2004 "Pig Feed Standard" issued by the Ministry of Agriculture and Rural Affairs of China in 2004 [25], as shown in Table 1.

**Table 1.** Nutrient formulation of basal feed for Bamei pigs.

| Material (*w/w* %) | | Nutrition Level | |
|---|---|---|---|
| Corn | 63.26 | Metabolizable energy (Kcal/kg) | 3095 |
| Wheat bran | 15 | Crude protein (%) | 14 |
| Soybean meal | 11.2 | Ca (%) | 0.55 |
| Alfalfa silage | 10 | Available phosphorus (%) | 0.21 |
| Rapeseed meal | 4 | Total phosphorus (%) | 0.46 |
| Soybean oil | 2.61 | Lysine (%) | 0.76 |
| Rape stalk | 0.85 | Methionine (%) | 0.21 |
| Stone powder | 1.1 | | |
| Dicalcium phosphate | 0.31 | | |
| Salt | 0.5 | | |
| Lysine | 0.17 | | |
| Premix | 1 | | |

Notes: 1. Nutritional composition of compound premix (content per kg): Vitamin A 160,000 IU, vitamin $D_3$ 50,000 IU, vitamin $E_1$ 500 mg, vitamin $K_3$ 80 mg, vitamin $B_1$ 45 mg, vitamin $B_2$ 110 mg, vitamin $B_6$ 80 mg, nicotinic acid 600 mg, pantothenic acid 300 mg, folic acid 10 ug, iron 4500 mg, copper 250 mg, iodine 50 mg, selenium 10 mg, calcium 17.5%, phosphorus 1.8%, lysine 5%, sodium chloride 9.5%, moisture <10%. 2. Nutritional levels were determined using near-infrared reflectance spectroscopy.

Fermented feed: 100 kg of basal feed supplemented with 50 kg of water and different bacterial agents or antibiotics according to the groups in Table 2, sealed in 100 L plastic buckets, and fermented for 7 days at room temperature (15–25 °C).

**Table 2.** Experimental group of five types of Bamei pig feed.

| Group | Feed Formula | Concentration of Additive |
|---|---|---|
| CK | basal feed | - |
| L | basal feed + *L. plantarum* QP28-1a | $1 \times 10^6$ CFU/g |
| B | basal feed + *B. subtilis* QB8a | $1 \times 10^6$ CFU/g |
| MIX | basal feed + *L. plantarum* QP28-1a and *B. subtilis* QB8a | $1 \times 10^6$ CFU/g (1:1) |
| A | basal feed + gentamycin (antibiotic) | 50 mg/kg |

### 2.2. Animals and Experimental Design

Experimental animals: Sixty healthy, newly weaned, 30-day-old Bamei ternary cross-bred pigs (Duroc × Long White × Bamei), half male and half female, with similar genetic backgrounds and an initial weight of about 10 kg, were randomly selected. This study was approved by the Life Sciences Ethics Review Committee of Zhengzhou University under certificate number ZZUIRB2021-111.

Experimental grouping: Sixty Bamei pigs were randomly divided into five groups of twelve pigs, each equally distributed in three pens according to Table 2. The first group was the control group (CK), fed the basal diet; the second group (L) was fed *L. plantarum* QP28-1a-fermented feed; the third group (B) was fed *B. subtilis* QB8a-fermented feed; the fourth group (MIX) was fed a mixed fermented feed, labeled as group MIX; and the fifth group (A) was fed antibiotic feed. The piglets were numbered, vaccinated, and dewormed.

Experimental design: We started to feed the weaned piglets after 7 days of fermentation and continued to feed them for 100 days until slaughter. The Bamei pigs were allowed to drink and feed freely and were fed four times a day at 8:00, 11:00, 15:00, and 19:00, with a small amount of feed remaining to measure the feed intake. The feed intake and residual feed were recorded accurately every day, and the pigs were weighed on the 0th, 30th, and 60th day. Nine fresh fecal samples (three fecal samples per pen) were collected from each treatment group on days 0, 10, 20, 30, 40, and 50 and stored in sterilized sampling tubes at

−80 °C for subsequent studies of microbial counts, microbial diversity and variability, and immunity. The pig house was disinfected and cleaned every day to keep it clean and dry and was naturally ventilated.

### 2.3. Number of Microorganisms in Feed

The number of culturable live bacteria (including yeast, Coliform bacteria, aerobic bacteria, *Clostridium*, *Bacillus*, and LAB) in the feed on days 2, 10, 20, 30, and 60 was analyzed via the culture method. The medium formulations and test steps were based on Zhang's method [26].

### 2.4. pH, Organic Acid, and Nutrient Content in Feed

Referring to the method of Zhao et al. [27], the pH values from days 2 to 60 were determined using a pH meter (Mettler-Toledo, GmbH, Griffin, Switzerland). Organic acids, including lactic, acetic, propionic, and butyric acids, were determined using high-performance liquid chromatography (Waters Alliance e2695, Waters, Milford, MA, USA). A Carbomix H-NP 10:8% (7.8 mm × 300 mm × 10 mm) column was used as the stationary phase at 55 °C. The mobile phase was 0.0254% sulfuric acid. The flow rate was set at 0.6 mL/min, the injection volume was 10 μL, and the detection wavelength of the UV detector (2489 UV, Waters, Milford, MA, USA) was 214 nm. Three replicates were randomly sampled from each treatment group's feed for testing.

The feeds were sampled on days 2, 10, 20, 30, 40, and 50 after fermentation and tested for changes in nutrient content across the five treatment groups. Three replicates were randomly sampled from each treatment group's feed for testing.

The dry matter content (DM) of the samples was determined using the AOAC standard method [28]. An analytical balance was used to accurately weigh 100 g samples of feed, recorded as M1, which were dried in a blast-drying oven (101-3A, Sunne, Shanghai, China) at 65 °C for 48 h. After removing the samples from the oven, they were cooled to room temperature and weighed, with the value recorded as M2. The formula for calculating the dry matter content was [1 − (M1 − M2)/M1] × 100%.

The crude protein (CP) content in the five feeds was determined using the Kjeldahl method, with reference to the Association of Official Analytical Chemists (AOAC) standards [29], employing a fully automated Kjeldahl meter (K9860, Shandong Haineng Future Technology Co., Jinan, China)

The neutral detergent fiber (NDF) and acid detergent fiber (ADF) content in the five feeds was determined and analyzed following the method described by Van Soest et al. [30].

The hemicellulose content was determined as follows: hemicellulose (%) = NDF (%) − ADF (%).

### 2.5. Determination of Growth, Slaughtering, and Immunity Performance of Bamei Pigs

The average daily gain (ADG), average daily feed intake (ADFI), and feed-conversion ratio (FCR) of each group were calculated based on the initial weight (day 0), final weight (day 60), and feed intake. All sixty pigs were tested.

Bamei pigs were fed for 100 days, fasted for 24 h, and then slaughtered. After bloodletting and removing the head, hooves, tail, and viscera, the left-side ketone body weight, backfat thickness, and eye-muscle area of the Bamei pigs were measured with reference to the National Agricultural Standard for the Determination of Carcass Traits of Lean Pigs [31] (NY/T 825-2004), issued by the Ministry of Agriculture of the People's Republic of China in 2004.

Nine immune-related cytokines, receptors, and adaptor proteins were selected as immune indicators to evaluate the effects of the different feeds on the immune performance of weaned piglets, including tumor necrosis factor α (TNF-α), interleukin 2 (IL-2), interleukin 1β (IL-1β), interferon γ (INF-γ), myeloid differentiation factor 88 (MyD88), Toll-like receptor 2 (TLR2), Toll-like receptor 4 (TLR4), interleukin 10 (IL-10) and nuclear factor kappa B (NF-KB). Methods and sampling: The production of these immune-related

factors in cecal samples on day 100 was measured using an ELISA kit (Beijing Dogesce Biotechnology Co., Ltd., Beijing, China) with reference to the method of Liu et al. [32]; five cecal samples were randomly selected for testing in each treatment group.

*2.6. Extraction of DNA and Sequencing of 16S rDNA from Fecal Samples*

The bacterial-community structure in the feces after 10 and 50 days of feeding with different diets was analyzed using a high-throughput-sequencing method, with three samples randomly selected from each treatment group for testing. Total bacterial DNA was extracted from Bamei pig fecal samples using a Bacterial DNA Kit D3350-02 (Omega Biotek, Norcross, GA, USA). The quality of the extracted DNA was determined using 1% agarose gel electrophoresis and a NanoDrop™ 2000 spectrophotometer (Thermo Fisher Scientific, Waltham, MA, USA). The 16S rDNA V3-V4 variable region of viable fecal samples was amplified by PCR using primers 338F (5′-ACTCCTACGGGGAGGCAGCAG-3′) and 806R (5′-GGACTACHVGGGTWTCTAAT-3′) with reference to the method of Wang et al. [33]. The amplification process was as follows: pre-denaturation at 95 °C for 3 min; denaturation at 95 °C for 30 s; annealing at 55 °C for 30 s; extension at 72 °C for 30 s, 30 cycles; extension at 72 °C for 10 min. The PCR products were purified, quantified, and then sequenced using Illumina's MiSeq PE300 platform (Shanghai Majorbio Bio-pharm Technology Co., Ltd., Shanghai, China).

*2.7. High-Throughput-Sequencing Analysis of Fecal Samples*

The raw sequences were quality controlled using Trimmomatic (version 0.36, USADELLAB.org, USA) and then spliced using FLASH (version 1.2.11, McKusick-Nathans Institute of Genetic Medicine, Baltimore, MD 21205, USA) to obtain the merged sequences, and low-quality sequences were removed using QIIME (version 1.9.1, the Knight and Caporaso labs, University of Colorado, USA) [34]. Sequences were clustered based on 97% sequence similarity using UPARSE (http://www.drive5.com/uparse/, accessed on 26 August 2023) to obtain operational taxonomic units (OTUs) [35]. Species classification was annotated for each sequence using RDP, and bacteria sequences were compared using the Silva database [36,37]. The alpha diversity index of the bacterial population was calculated using Mothur (https://www.mothur.org/wiki/Download_mothur, accessed on 11 October 2023). UPGMA sample clustering trees were constructed using QIIME. The beta diversity and Pearson correlation heatmap of the bacterial population with environmental factors were determined based on a Kruskal–Wallis test using R (version 2.15.3, Lucent Technologies, New Jersey, USA). Linear discriminant analysis (LEfSe) was performed using Python software (version 2.7.14, Python Software Foundation, Delaware, USA).

*2.8. Statistical Analysis*

All collected data were recorded, organized, and calculated using Excel 2018 (Microsoft, Washington, DC, USA). The experimental data were subjected to one-way ANOVA using SPSS 22.0 (IBM Inc., Armonk, NY, USA), and significant differences between treatments were statistically analyzed using Tukey's post hoc multiple comparison method. $p < 0.05$ and $p < 0.01$ indicate significant differences and highly significant differences, respectively, while $p > 0.05$ indicates no significant difference.

## 3. Results

*3.1. Effect of Adding Probiotics or Antibiotics on the Microbiota of Feeds*

The result of microbial colony count of the fermented feeds is shown in Table 3. The type of treatment significantly affected the number of yeast, Coliform bacteria, aerobic bacteria, *Clostridium*, *Bacillus*, and LAB in the five feeds throughout the experiment ($p < 0.01$). The number of LAB was significantly higher in groups L and MIX on the second day ($p < 0.01$), reaching 6.10 log cfu/g and 6.17 log cfu/g, respectively, whereas the number of *Bacillus* in groups B and MIX was significantly higher than in groups CK and A, reaching 5.89 log cfu/g and 5.64 log cfu/g, respectively. The amount of yeast in groups B and MIX

was significantly higher than in the other groups ($p < 0.05$), and the number of Coliform bacteria in group MIX was significantly lower than in the other groups. The growth of aerobic bacteria was significantly inhibited by the addition of either the probiotic *L. plantarum* QP28-1a or antibiotics, and the number of *Clostridium* in group L was also significantly lower than in the other groups ($p < 0.05$). On day 10, the number of LAB in groups L, B, and MIX was significantly higher; the number of *Bacillus* in group MIX was significantly lower ($p < 0.05$); the number of *Clostridium* in group MIX was significantly lower than in the other groups ($p < 0.05$); and the number of aerobic bacteria in group MIX was significantly lower than in the other groups ($p < 0.05$). Notably, the number of Coliform bacteria and aerobic bacteria in group L was significantly lower than in the other groups ($p < 0.05$), indicating that the addition of strain *L. plantarum* QP28-1a more effectively inhibited the growth of these bacteria than the antibiotic group. Additionally, the number of *Clostridium* and aerobic bacteria in group MIX on day 10 was significantly lower than in the other groups ($p < 0.05$). On the 20th day, the number of LAB in group B was significantly higher; the number of *Bacillus* in group CK was significantly higher than in the other groups; and the growth of *Coliform* bacteria, aerobic bacteria, *Clostridium*, and *Bacillus* was effectively inhibited in groups L and MIX. The number of microorganisms did not change significantly between days 30 and 20, and the growth of Coliform bacteria, aerobic bacteria, *Clostridium*, and *Bacillus* was still effectively inhibited in groups L and MIX. On the 60th day, the number of LAB and *Bacillus* in groups L and MIX was significantly lower ($p < 0.05$), and the number of LAB and aerobic bacteria in group B was significantly higher ($p < 0.05$) than in the other groups, reaching 8.66 log cfu/g and 8.46 log cfu/g, respectively, whereas the growth of Coliform bacteria, aerobic bacteria, and *Clostridium* in groups L and MIX still displayed a significant inhibitory effect ($p < 0.05$), and the bacteriostatic effect was stronger than in the antibiotic group.

**Table 3.** Results of microbial colony counts of five types of Bamei pig feeds in 60 days.

| Time (Day) | Microbial Colony Counts log cfu/g | Treatment Groups | | | | | SEM | *p* |
|---|---|---|---|---|---|---|---|---|
| | | CK | L | B | MIX | A | | |
| 2 | *Yeast* | ND | ND | 3.66 [a] | 4.05 [a] | ND | 0.18 | ** |
| | *Coliform bacteria* | 6.57 [a] | 6.08 [a] | 6.24 [a] | 5.34 [b] | 6.09 [a] | 0.11 | ** |
| | *Aerobic bacteria* | 6.66 [a] | 6.07 [c] | 6.67 [a] | 5.67 [d] | 6.48 [b] | 0.04 | ** |
| | *Clostridium* | 5.54 [a] | 5.67 [a] | 5.71 [a] | 4.68 [b] | 5.41 [a] | 0.08 | ** |
| | *Bacilli* | 5.34 [b] | 5.39 [a] | 5.89 [a] | 5.64 [a] | 4.91 [c] | 0.09 | ** |
| | LAB | 5.56 [c] | 6.10 [a] | 5.77 [b] | 6.17 [a] | 5.65 [bccc] | 0.04 | ** |
| 10 | *Yeast* | ND | ND | 4.15 [a] | 3.82 [b] | ND | 0.03 | ** |
| | *Coliform bacteria* | 6.57 [a] | 4.29 [d] | 6.07 [b] | 4.61 [c] | 6.29 [a] | 0.04 | ** |
| | *Aerobic bacteria* | 6.61 [a] | 5.99 [b] | 6.75 [a] | 5.83 [b] | 6.52 [a] | 0.06 | ** |
| | *Clostridium* | 5.43 [a] | 5.14 [a] | 5.24 [a] | 4.21 [b] | 5.40 [a] | 0.08 | ** |
| | *Bacilli* | 5.57 [a] | 5.46 [a] | 5.47 [a] | 4.59 [b] | 5.21 [ab] | 0.13 | ** |
| | LAB | 5.43 [c] | 6.13 [b] | 7.71 [a] | 6.12 [b] | 5.91 [b] | 0.03 | ** |
| 20 | Yeast | ND | ND | 4.65 | ND | ND | 0.01 | ** |
| | *Coliform bacteria* | 6.56 [a] | 4.20 [e] | 4.63 [c] | 4.35 [d] | 6.20 [b] | 0.02 | ** |
| | *Aerobic bacteria* | 6.75 [b] | 4.94 [d] | 8.60 [a] | 4.58 [e] | 4.89 [c] | 0.04 | ** |
| | *Clostridium* | 5.31 [a] | 5.46 [a] | 4.94 [b] | 4.23 [c] | 4.89 [b] | 0.08 | ** |
| | *Bacilli* | 5.37 [a] | 4.76 [c] | 4.75 [c] | 4.98 [b] | 4.99 [b] | 0.05 | ** |
| | LAB | 5.25 [c] | 6.04 [b] | 8.03 [a] | 6.14 [b] | 5.20 [c] | 0.07 | ** |
| 30 | Yeast | ND | ND | 4.06 | ND | ND | 0.02 | ** |
| | *Coliform bacteria* | 6.63 [a] | 4.64 [b] | 4.42 [b] | 4.12 [b] | 6.17 [a] | 0.18 | ** |
| | *Aerobic bacteria* | 6.57 [b] | 4.71 [c] | 8.26 [a] | 4.48 [c] | 6.40 [b] | 0.07 | ** |
| | *Clostridium* | 5.51 [a] | 4.93 [b] | 5.53 [a] | 3.54 [c] | 5.04 [b] | 0.06 | ** |
| | *Bacilli* | 5.31 [b] | 3.95 [c] | 5.63 [a] | 3.75 [c] | 5.24 [b] | 0.07 | ** |
| | LAB | 5.42 [c] | 6.11 [b] | 8.81 [a] | 5.96 [b] | 5.31 [c] | 0.10 | ** |

**Table 3.** *Cont.*

| Time (Day) | Microbial Colony Counts log cfu/g | Treatment Groups | | | | | SEM | *p* |
|---|---|---|---|---|---|---|---|---|
| | | CK | L | B | MIX | A | | |
| 60 | *Yeast* | ND | ND | ND | ND | ND | - | - |
| | *Coliform bacteria* | 6.40 [a] | 4.05 [b] | 6.01 [a] | 4.14 [b] | 6.05 [a] | 0.10 | ** |
| | *Aerobic bacteria* | 6.72 [b] | 4.59 [d] | 8.46 [a] | 4.40 [e] | 6.52 [c] | 0.04 | ** |
| | *Clostridium* | 4.77 [b] | 3.82 [c] | 5.88 [a] | 4.03 [c] | 4.91 [b] | 0.14 | ** |
| | *Bacilli* | 5.01 [b] | 3.95 [c] | 5.66 [a] | 4.69 [b] | 4.97 [b] | 0.15 | ** |
| | LAB | 5.43 [b] | 4.37 [c] | 8.66 [a] | 4.32 [c] | 5.37 [b] | 0.08 | ** |

Notes: 1. CK, basal feed group; L, fermented feed group with the addition of *L. plantarum* QP28-1a; B, fermented feed group with the addition of *B. subtilis* QB8a; MIX, fermented feed group with the addition of *L. plantarum* QP28-1a and *B. subtilis* QB8a; A, feed group with antibiotics feed. 2. Values are means of three parallel groups. Different lowercase letters for the same indicator in the same row indicate significant differences ($p < 0.05$). ** $p < 0.01$. SEM, standard error; ND, not detected.

*3.2. pH and Organic Acid Content of Different Bamei Pig Feeds*

The pH and organic acid content of the different feeds were determined over a period of 60 days. The dynamics of the pH are shown in Figure 1A. The pH of the CK group did not change significantly ($p > 0.05$) over the 60 days, remaining at around 6.1. The pH of group A, with antibiotic supplementation, showed a decreasing and then increasing trend, reaching a minimum value of 5.69 at day 40. The pH of group L, with added potentially probiotic *L. plantarum* QP28-1a, dropped sharply to 5.52 ($p < 0.05$) on the second day of fermentation. The pH of group B, with added potentially probiotic *B. subtilis* QB8a, and group MIX also decreased significantly ($p < 0.05$). Specifically, the pH of group MIX decreased to 4.53 at day 30, the lowest pH value of all observations ($p < 0.05$), whereas the pH of group B increased to 5.58 at day 40. Regarding the changes in organic-acid content, as shown in Figure 1B. Low levels of lactic acid were detected in all treatment groups on day 2, with groups L and B having significantly higher levels than the other groups ($p < 0.05$). By day 20, the highest level of lactic acid was detected in group B at 65.31 g/kg DM ($p < 0.05$). Acetic acid was detected in groups L, B, and MIX, with the highest level in group MIX at 29.93 g/kg DM ($p < 0.05$). By day 60, group L had the highest level of lactic acid at 58.95 g/kg DM ($p < 0.05$).

*3.3. Differences in Nutrient Content in Different Feeds*

The nutrient composition of the five Bamei pig feeds at different fermentation stages is shown in Table 4. The dry-matter content of groups CK and A was significantly higher than that of the other groups at all stages ($p < 0.05$). Group L had the lowest average-dry-matter content throughout fermentation ($p < 0.05$). The dry-matter content of groups B and MIX tended to increase with fermentation duration. The crude protein measurements showed that the content in groups L and B was significantly higher than in the other groups at 10 days, and the content in groups CK and A did not change significantly throughout fermentation. The average crude protein content in the feeds of group L was significantly higher than that of the other groups throughout fermentation ($p < 0.05$). The content reached its peak on the 10th day, presenting a value significantly higher than on day 2 ($p < 0.05$), indicating that the addition of *L. plantarum* QP28-1a significantly increased the crude protein content of fermented feed.

The addition of *Bacillus subtilis* QB8a significantly ($p < 0.05$) reduced the NDF content of the feeds in groups B and MIX. With an increase in the fermentation time, the NDF content in groups L, B, and MIX tended to decrease, and that in group MIX reached the lowest value of 26.88% on the 40th day, significantly lower than the other treatment groups ($p < 0.05$).

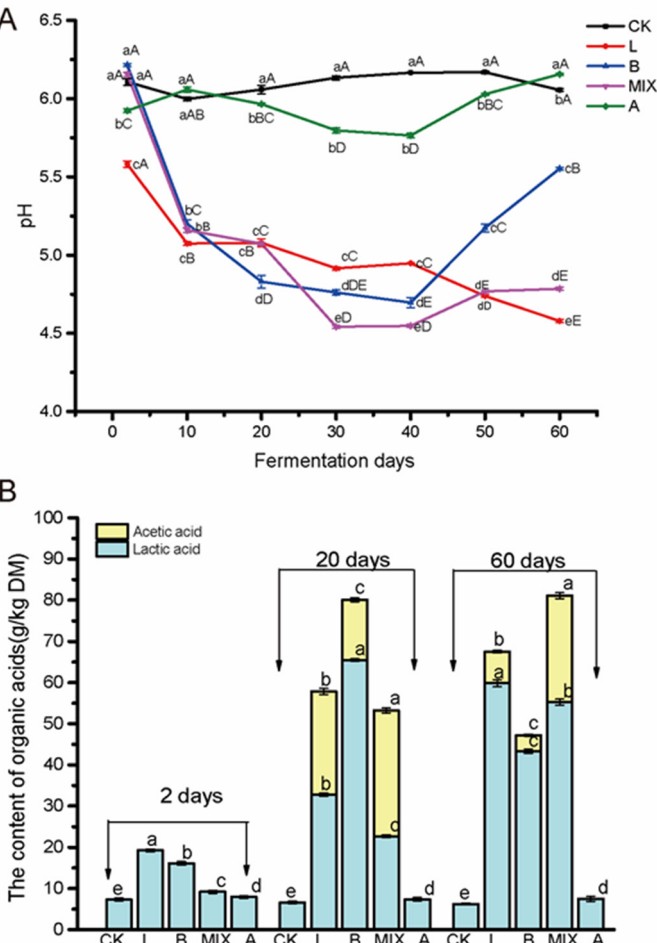

**Figure 1.** Dynamics of (**A**) pH and (**B**) organic acids in different treatment groups of feeds in 60 days. Values are means of three parallel experiments. Different lowercase letters for the same indicator indicate significant differences between different treatment groups ($p < 0.05$), and different capital letters in the same treatment group indicate significant differences in pH between different days ($p < 0.05$).

The ADF content of groups L, B, and MIX with *L. plantarum* QP28-1a or *B. subtilis* QB8a was significantly lower than that of groups CK and A ($p < 0.05$), and that of group L decreased significantly with the increase in fermentation time ($p < 0.05$).

The average hemicellulose content in groups B and MIX was significantly lower than in the other groups ($p < 0.05$), while it did not differ significantly between groups L and CK, and A. This indicated that the addition of *B. subtilis* QB8a significantly reduced the hemicellulose content in the fermented feeds.

### 3.4. Effect of Fermented Feeds on Growth and Slaughter Performance of Bamei Pigs

The growth and slaughter performance of the treatment groups of Bamei pigs are shown in Table 5. The differences in initial weight, final weight, and average daily gain were not significant ($p > 0.05$) among groups CK, L, B, MIX, and A. However, the ADFI of groups B and MIX was significantly higher than that of group L. The FCR of group B was the highest at 3.21%, and that of group MIX was significantly lower than that of the other groups ($p < 0.05$). The backfat thickness and left-side ketone body mass of the Bamei pigs did not differ significantly between treatment groups ($p > 0.05$), and the eye-muscle area of the pigs in groups CK and MIX was significantly higher than that in the other groups ($p < 0.05$).

**Table 4.** Nutrient composition (dry-matter basis) of five types of pig feeds for Bamei pig.

| Nutrient Composition | Treatment Groups | Fermented Days (Day) | | | | | | Average Value | SEM | p Value | | |
|---|---|---|---|---|---|---|---|---|---|---|---|---|
| | | 2 | 10 | 20 | 30 | 40 | 60 | | | T | D | T × D |
| DM (%) | CK | 88.30 a | 88.07 a | 87.81 a | 87.27 a | 87.59 a | 88.31 a | 87.89a | 0.03 | ** | ** | ** |
| | L | 54.39 b | 54.77 bc | 55.28 bc | 55.31 c | 55.59 c | 55.70 c | 55.17 c | | | | |
| | B | 54.77 bC | 55.53 bBC | 56.73 bB | 58.30 bA | 58.95 bA | 58.51 bA | 57.13 b | | | | |
| | MIX | 53.47 bB | 53.59 cB | 53.83 cB | 54.77 cAB | 55.65 cA | 54.93 cAB | 54.37 d | | | | |
| | A | 87.80 a | 88.20 a | 87.57 a | 86.57 a | 87.73 a | 88.17 a | 87.67 a | | | | |
| CP (%DM) | CK | 12.90 | 12.31 c | 13.33 ab | 13.20 ab | 13.19 ab | 12.84 b | 12.96 b | 0.13 | ** | ** | ** |
| | L | 12.28 B | 14.21 aA | 13.01 aA | 13.44 aA | 13.71 aA | 13.88 aA | 13.58 a | | | | |
| | B | 12.50 B | 13.62 abA | 12.80 bAB | 12.93 abAB | 12.81 bAB | 12.56 bB | 12.87 b | | | | |
| | MIX | 12.52 AB | 13.31 bA | 12.72 bAB | 12.36 bB | 12.41 bAB | 12.99 bAB | 12.71 b | | | | |
| | A | 12.31 | 12.75 bc | 12.74 b | 13.24 a | 12.85 ab | 12.69 b | 12.77 b | | | | |
| NDF (%DM) | CK | 30.64 | 30.63 b | 31.07 a | 31.55 ab | 31.17 a | 31.48 a | 31.09 a | 0.38 | ** | ** | ** |
| | L | 31.87 AB | 33.25 aA | 31.03 aAB | 30.75 abAB | 30.56 abB | 29.81 abB | 31.03 a | | | | |
| | B | 30.70 AB | 32.20 abA | 28.40 bBC | 28.01 cC | 28.84 bcBC | 28.40 bcBC | 29.43 b | | | | |
| | MIX | 31.59 A | 30.42 bA | 29.53 abAB | 29.09 bcABC | 26.88 cC | 27.12 cBC | 29.11 b | | | | |
| | A | 31.73 | 31.32 ab | 31.82 a | 31.92 a | 31.80 a | 31.76 a | 31.72 a | | | | |
| ADF (%DM) | CK | 13.54 a | 13.48 a | 13.95 a | 13.79 a | 13.66 a | 13.55 a | 13.66 a | 0.09 | ** | * | ** |
| | L | 12.82 bcA | 12.77 bA | 12.07 bB | 12.46 bAB | 12.09 bB | 12.29 bAB | 12.42 c | | | | |
| | B | 13.28 abA | 13.51 aA | 12.56 bB | 12.66 bB | 13.23 aAB | 13.29 aA | 13.09 b | | | | |
| | MIX | 12.66 c | 12.45 b | 12.40 b | 12.25 b | 12.29 b | 12.06 b | 12.35 c | | | | |
| | A | 13.57 a | 13.56 a | 13.65 a | 13.81 a | 13.69 a | 13.52 a | 13.64 a | | | | |
| Hemicellulose (%DM) | CK | 18.09 ab | 17.36 b | 17.12 ab | 17.76 a | 17.52 a | 17.93 a | 18.03 a | 0.35 | ** | ** | ** |
| | L | 19.05 a | 19.48 a | 19.08 a | 18.30 a | 18.47 a | 17.51 a | 18.62 a | | | | |
| | B | 16.58 bA | 17.83 bA | 15.04 bB | 14.53 bB | 14.77 bB | 14.27 bB | 15.50 c | | | | |
| | MIX | 18.92 aA | 17.97 bA | 17.13 abA | 16.84 aA | 14.59 bB | 15.06 bB | 16.75 b | | | | |
| | A | 18.16 ab | 17.75 b | 18.17 a | 18.11 a | 18.11 a | 18.22 a | 18.09 a | | | | |

Notes: 1. DM, dry matter content; CP, crude protein; NDF, neutral detergent fiber; ADF, acid detergent fiber. 2. CK, basal feed group; L, group of fermented feed with the addition of *L. plantarum* QP28-1a; B, group of fermented feed with the addition of *B. subtilis* QB8a; MIX, group of composite fermented feed with *L. plantarum* QP28-1a and *B. subtilis* QB8a; A, feed group with antibiotic feed. 3. Values are means of three parallel experiments; different lowercase letters in the same column indicate significant differences between treatment groups ($p < 0.05$), and different capital letters in the same row indicate significant differences in different days for this indicator ($p < 0.05$). SEM, standard error; T, treatment group; D, days of forage fermentation; T × D, interaction effect of treatment group and days of forage fermentation. * $p < 0.05$; ** $p < 0.01$.

**Table 5.** Effect of fermented feeds on growth and slaughter performance of Bamei pigs.

| Item | Dietary Treatments | | | | | SEM | *p*-Value |
|---|---|---|---|---|---|---|---|
| | **CK** | **L** | **B** | **MIX** | **A** | | |
| Initial weight (kg) | 9.12 | 8.88 | 9.04 | 8.87 | 8.83 | 0.11 | NS |
| Final weight (kg) | 29.37 | 28.65 | 29.74 | 31.83 | 30.06 | 1.07 | NS |
| ADG (kg/d) | 0.337 | 0.329 | 0.345 | 0.383 | 0.354 | 0.02 | NS |
| ADFI (kg/d) | 1.05 [ab] | 0.99 [b] | 1.13 [a] | 1.14 [a] | 1.09 [ab] | 0.02 | * |
| FCR | 3.10 [b] | 3.02 [bc] | 3.21 [a] | 2.99 [c] | 3.07 [bc] | 0.02 | * |
| Left ketone weight (kg) | 15.72 | 14.18 | 13.59 | 15.04 | 15.13 | 0.17 | NS |
| Backfat thickness(mm) | 16.7 | 16.03 | 16.59 | 17.52 | 16.64 | 0.29 | NS |
| Eye-muscle area (cm$^2$) | 26.45 [a] | 20.57 [b] | 18.14 [b] | 25.01 [a] | 21.45 [b] | 0.57 | * |

Note: 1. Different lowercase letters in the same row indicate significant differences ($p < 0.05$); *, $p < 0.05$; NS, not significantly different. Data are means of 12 pigs. SEM, standard error of the mean. 2. ADG, average daily gain; ADFI, average daily feed intake; FCR, feed-conversion ratio. 3. CK, basal feed group; L, group of fermented feed with the addition of *L. plantarum* QP28-1a; B, group of fermented feed with the addition of *B. subtilis* QB8a; MIX, group of mixed fermented feed with *L. plantarum* QP28-1a and *B. subtilis* QB8a; A, group of antibiotic feed.

### 3.5. Effect of Fermented Feeds on the Immune Performance of Bamei Pigs

The production of immune factors in the treatment groups of Bamei pigs after feeding for 100 days is shown in Figure 2. The production of immune factors IL-2, IL-1β, MyD88, TLR2, TLR4, IL-10, and NF-KB differed significantly between the groups CK, L, B, MIX, and A ($p < 0.05$). The production of IL-2 and IL-10 in groups L, B, MIX, and A was significantly higher than in group CK ($p < 0.05$); the production of immunity factors IL-1β, MyD88, TLR4, and NF-KB in group CK was significantly higher than in groups L, B, MIX, and A ($p < 0.05$); and the production of immune factor TLR2 in group CK was significantly higher than in groups B, MIX, and A ($p < 0.05$).

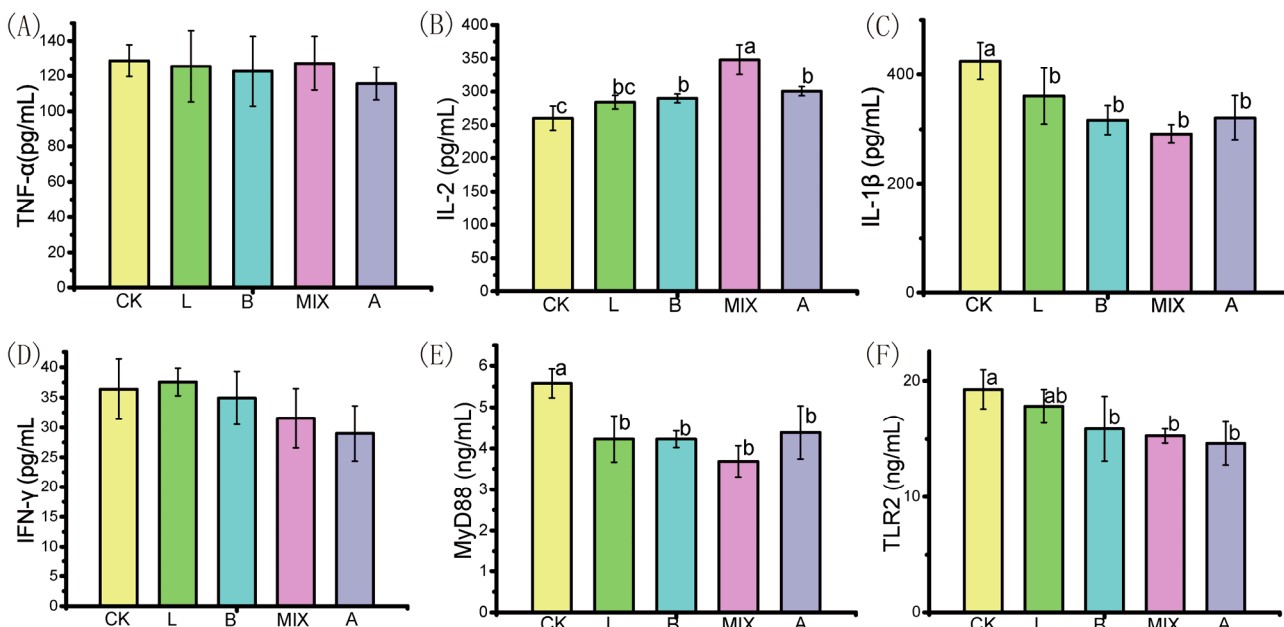

**Figure 2.** *Cont*.

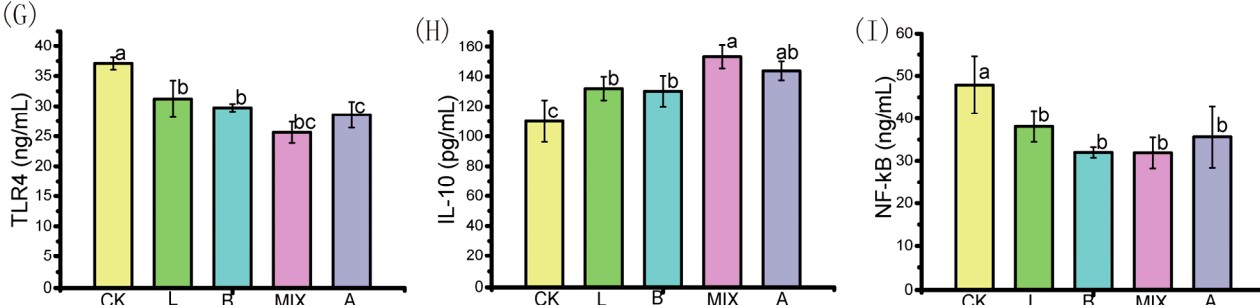

**Figure 2.** Production of intestinal immune factors after 50 days of feeding Bamei pigs. (**A**) TNF-α; (**B**) IL-2; (**C**) IL-1β; (**D**) INF-γ; (**E**) MyD88; (**F**) TLR2; (**G**) TLR4; (**H**) IL-10; (**I**) NF-KB. Different lowercase letters for the same indicator indicate significant differences between different treatment groups ($p < 0.05$).

*3.6. Effect of Fermented Feeds on Intestinal Microorganisms of Bamei Pigs*

3.6.1. Effect of Fermented Feeds on the Diversity of Intestinal Bacteria in Bamei Pigs

The bacterial-community structure in the feces of the different treatment groups at 10 and 50 days was analyzed using 16S rDNA high-throughput sequencing. The α-diversity of the bacterial communities was expressed using the Chao and Shannon indices, which represent richness and diversity, respectively. As shown in Figure 3, after 10 days of fermentation, the Chao and Shannon indices were significantly higher ($p < 0.05$) in groups CK, L, B, and MIX than in group A. By the 50th day, the Shannon index was significantly higher ($p < 0.05$) in groups L, MIX, and A than in group CK.

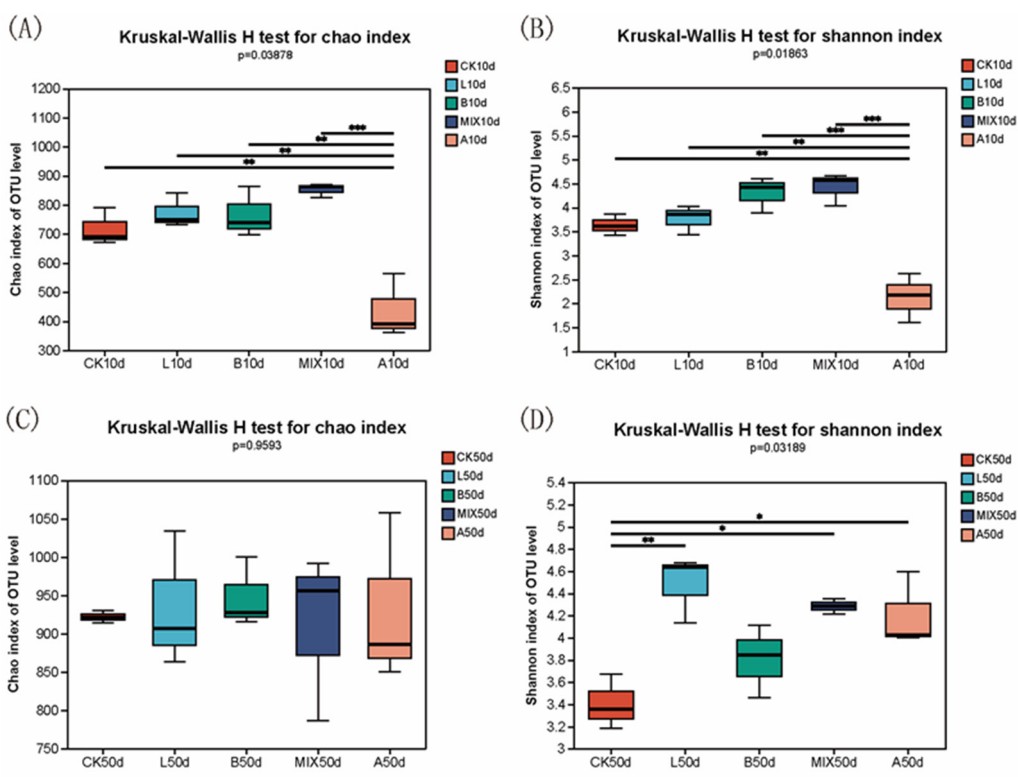

**Figure 3.** α-diversity of bacterial communities at 10 days and 50 days of feeding Bamei pigs: (**A**) 10 days Chao index; (**B**) 10 days Shannon index; (**C**) 50 days Chao index; (**D**) 50 days Shannon index. * $p < 0.05$; ** $p < 0.01$; *** $p < 0.001$.

The β-diversity of the intestinal bacterial microbiota of the Bamei pigs was analyzed using sample-level cluster analysis and principal coordinates analysis (PCoA), as presented

in Figure 4. Samples from the same treatment group were clustered together and had similar bacterial-abundance structures at the genus level. The relative abundance of *Lactobacillus* was significantly higher in group A than in the other treatment groups on day 10. At day 50, the relative abundance of *Clostridium* was higher in group B, and that of *Streptococcus* was higher in group CK. Clear distinctions in bacterial-community structure were observed between treatment groups, as well as the aggregation of samples within treatment groups (Figure 4B,C), suggesting that the addition of either a potential probiotic (*L. plantarum* QP28-1a or *B. subtilis* QB8a) or antibiotics significantly altered the bacterial structure of the intestinal microbiota.

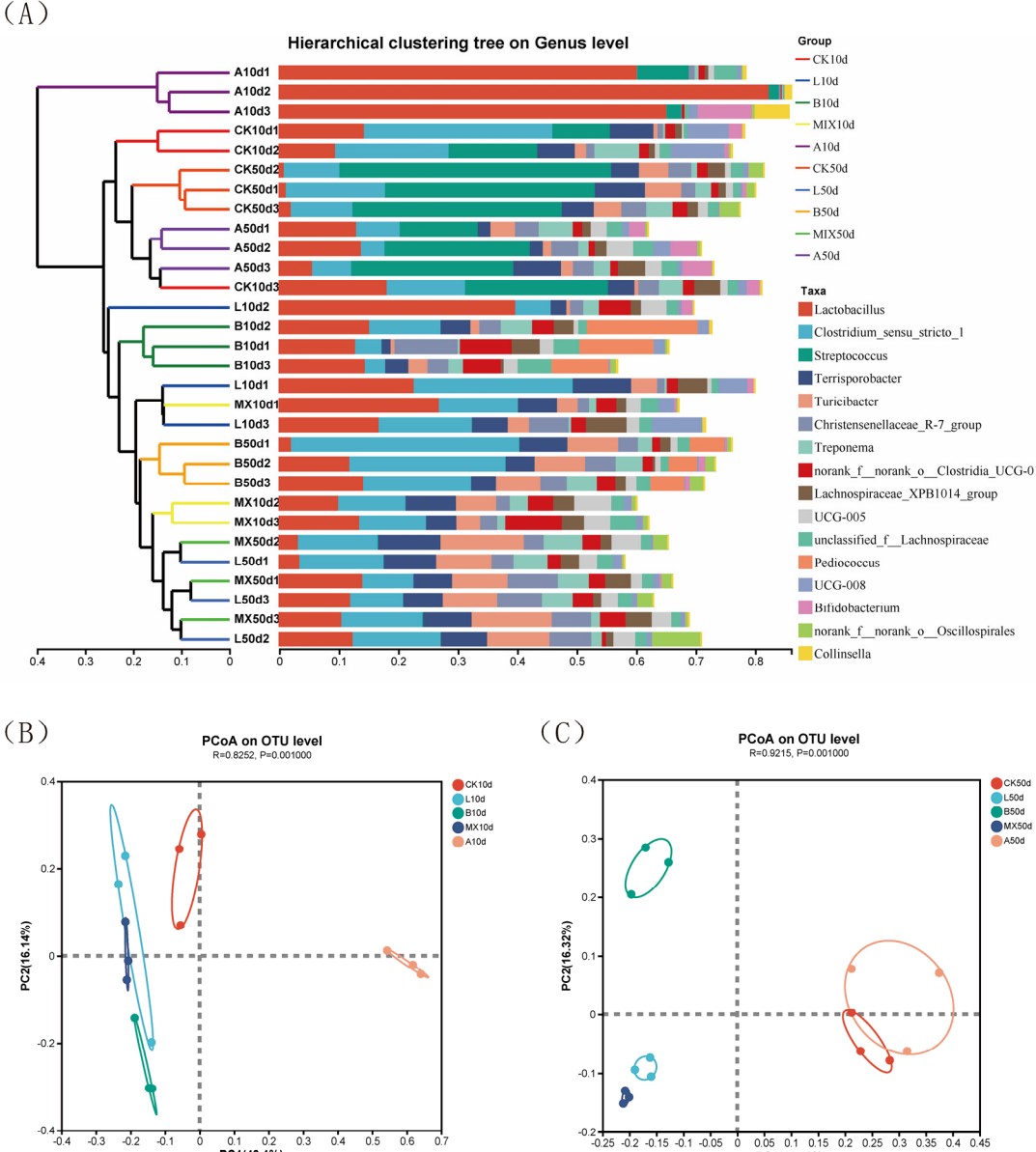

**Figure 4.** Analysis of β-diversity of fecal samples from Bamei pigs. (**A**) Sample clustering analysis on genus level; PCoA analysis of bacterial-community compositions at 10 days (**B**) and 50 days (**C**).

### 3.6.2. Comparison of the Composition of Intestinal Bacterial Community

The bacterial composition of the Bamei pigs' feces was analyzed on days 10 and 50 (Figure 5). At the phylum level, Firmicutes was consistently the dominant phylum on days 10 and 50. On day 10, the relative abundance of Bacteroidota was higher in groups B and MIX than in the other groups by 5.7% and 6.3%, respectively, and that of

Actinobacteriota (6.7%) in group A was higher than in the other groups. On day 50, the relative abundance of Firmicutes in group L (81.9%) was lower than in group CK (89.7%), and the relative abundance of Bacteroidota in group L (9.5%) was higher than that in group CK (3.6%). At the genus level (Figure 5C,D), the relative abundance of *Lactobacillus* on day 10 increased in groups L (26.4%) and MIX (16.8%) compared to group CK (14.0%); the relative abundance of *Clostridium* was significantly lower; and that of *Streptococcus* virtually disappeared. The relative abundance of *Pediococcus* increased in group B (13.6%) compared to the other groups. Notably, the relative abundance of *Lactobacillus* in group A, with added antibiotics, was significantly higher than in the other treatments groups, reaching 69.3%, with *Lactobacillus* and *Streptococcus* taking a competitive advantage. By day 50, the relative abundance of *Lactobacillus* in groups L, B, A, and MIX was 9.5%, 9.4%, 9.3%, and 10.9%, respectively, higher than in group CK (1.4%). Meanwhile, the relative abundance of *Streptococcus* in groups L, B, and MIX was significantly lower than in groups CK and A. *Lactobacillus* was no longer absolutely dominant in group A (10.9%) but formed a dominant group with *Streptococcus* (21.6%) and *Clostridium* (5.9%).

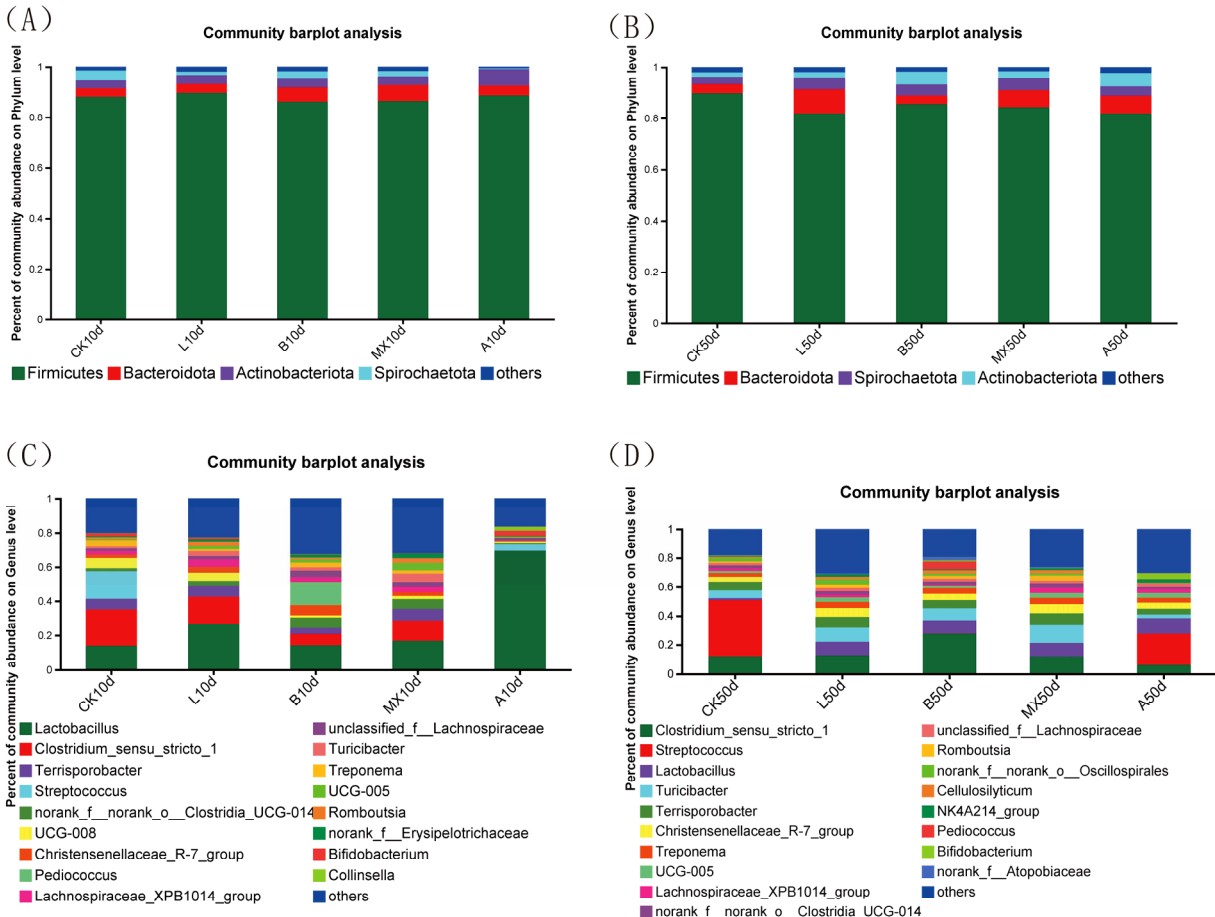

**Figure 5.** Bacterial-relative-abundance analysis on (**A**) phylum level at day 10; (**B**) phylum level at day 50; (**C**) genus level at day 10; (**D**) genus level at day 50.

Differences in the bacterial microbiota composition among treatment groups were further explored using linear discriminant analysis (LEfSe) at genus level. On day 10 (Figure 6A), *Clostridium* and *Streptococcus* were significantly enriched in group CK; *Oscillospiraceae* was significantly enriched in group L; *Pediococcus*, *Clostridia*, and *Christensenellaceae* were significantly enriched in group B; *Terrisporobacter*, *Turicibacter*, *Romboutsia*, and *Erysipelotrichaceae* were significantly enriched in group MIX; and *Collinsella* and *Prevotella* were most abundant in group A. By day 50 (Figure 6B), *Streptococcus* and *Microbacterium*

were significantly enriched in group CK; *Prevotella* was significantly enriched in group L; *Clostridium* and *Pediococcus* were significantly enriched in group B; *Turicibacter*, *Romboutsia*, and *Erysipelotrichaceae* were significantly enriched in group MIX; and *Bifidobacterium*, *Catenibacterium*, and *Oscillospira* were significantly enriched in group A.

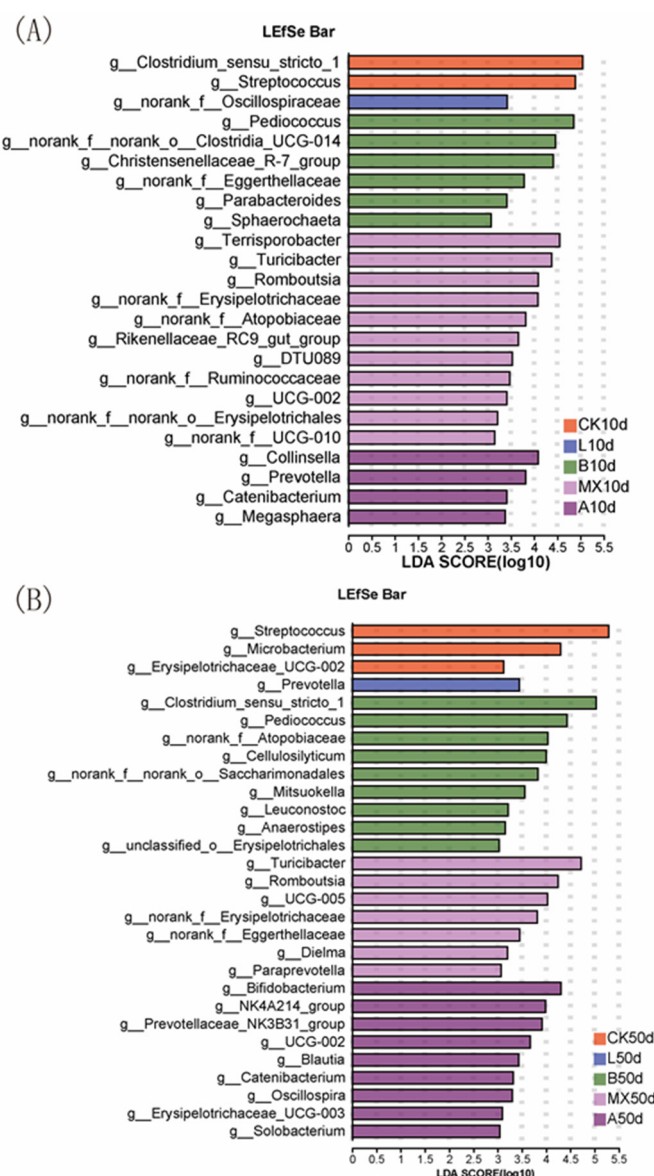

**Figure 6.** Comparison of microbial variations using LEfSe on genus level at (**A**) day 10, and (**B**) day 50.

### 3.7. Analysis of the Correlation between Intestinal-Bacterial-Community Structure and Immune Cytokines

The correlation between the relative abundance of bacteria and immune-related cytokines in cecal samples of Bamei pigs was evaluated using Pearson correlation analysis (Figure 7). The relative abundance of *Lactobacillus* was positively correlated with IL-2; significantly positively correlated with IL-10 ($r = 0.53$, $p < 0.05$); and significantly negatively correlated with MyD88 ($r = -0.62$, $p < 0.05$), NF-KB ($r = -0.57$, $p < 0.05$), TLR2 ($r = -0.61$, $p < 0.05$), and INF-γ ($r = -0.68$, $p < 0.01$). The relative abundance of *Erysipelotrichaceae* showed a highly significant positive correlation with IL-2 ($r = 0.67$, $p < 0.01$) and IL-10 ($r = 0.76$, $p < 0.01$) and a highly significant negative correlation with MyD88 ($r = -0.78$, $p < 0.01$), NF-KB ($r = -0.79$, $p < 0.01$), TLR4 ($r = -0.80$, $p < 0.01$), and IL-1β ($r = -0.61$, $p < 0.01$). The relative abundance of *Streptococcus* showed a highly significant negative

correlation with IL-10 (r = −0.66, *p* < 0.01); a very significant negative correlation with MyD88 (r = 0.73, *p* < 0.01), NF-KB (r = 0.89, *p* < 0.01), and TLR4 (r = 0.65, *p* < 0.01); and a significant positive correlation with IL-1β (r = 0.55, *p* < 0.01).

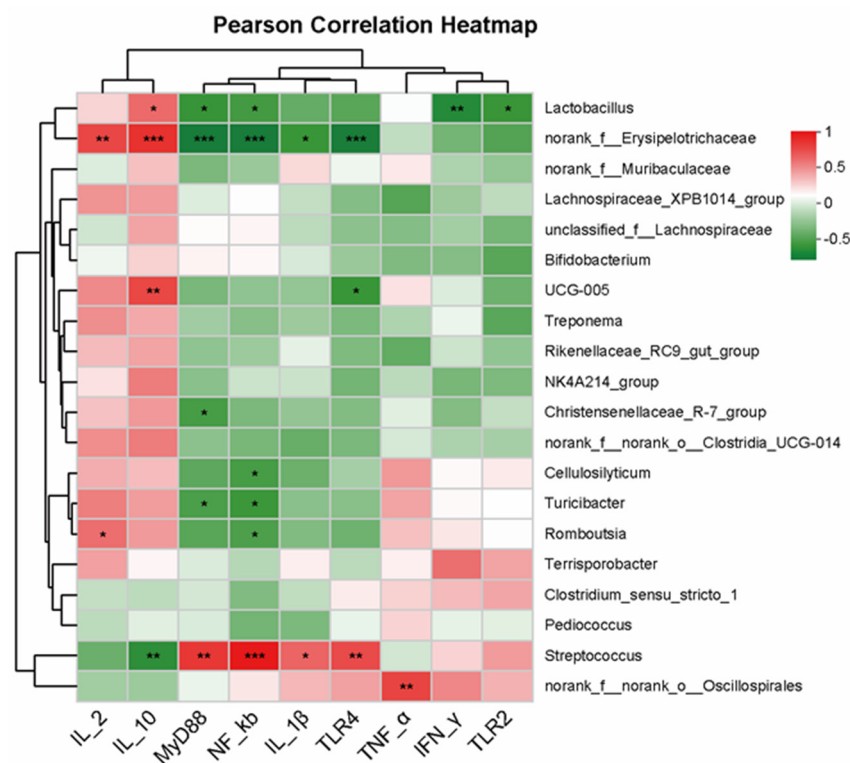

**Figure 7.** Pearson correlation heatmap analysis of relative abundance of bacteria and immune-related cytokines in fecal samples of Bamei pigs at day 50. * *p* < 0.05; ** *p* <0.01; *** *p* < 0.001.

## 4. Discussion

In recent years, the application of probiotics in farming has been extensively promoted due to the total ban on antibiotics in this industry. Although probiotics have the advantages of being healthy and non-residual, antibiotics are low cost, rapidly reduce piglet mortality and diarrhea rates, and improve growth performance in pig farming [2,38]. In this study, the effects and benefits of providing potentially probiotic *L. plantarum*- and *B. subtilis*-fermented feed to newly weaned Bamei piglets were investigated in four dimensions—feed quality, growth performance, gut microbiota, and immunity—to lay a working foundation for antibiotic replacements in the breeding of Bamei pigs.

After adding potentially probiotic *L. plantarum* QP28-1a or *B. subtilis* QB8a for fermentation, the pH of groups L, B, and MIX was significantly reduced (*p* < 0.05), and the fermentation process was accompanied by the production of lactic acid and acetic acid. The lower pH and the production of organic acids, which kept the feeds in an acidic state for a long period of time, inhibited the growth of harmful bacteria, thus prolonging the feed's preservation. A lower pH and higher lactic acid concentration are important for the antimicrobial properties of fermented feeds, which not only effectively inhibit the proliferation and colonization of pathogens, but also facilitate the formation of the intestinal barrier [39,40]. Tajima et al. [41] used *L. plantarum* for the fermentation of liquid feeds, and after 18 h the lactic acid content of the feed reached 237 mmol/g and the pH was reduced to 3.9, which increased the acidity of the animals' gastrointestinal tracts and their digestive enzyme activity. Fermented feeds supplemented with *L. plantarum* QP28-1a or *B. subtilis* QB8a had reduced NDF and ADF content and increased nutritional value, improving the use of roughage by animals. Similarly, Tabacco et al. [42] found that the addition of *Lactobacillus buchneri* significantly reduced the NDF and ADF content of whole-plant corn silage and improved the nutrition and quality of the feed. Only groups B and MIX, with the

addition of *B. subtilis* QB8a, presented reduced hemicellulose content, suggesting that the two strains differ in their ability to hydrolyze various types of cellulose, with *L. plantarum* QP28-1a probably lacking the enzyme for hydrolyzing hemicellulose.

*Lactobacillus* and *Bacillus* subtilis can produce active digestive enzymes such as lipase, protease, and amylase; metabolites such as organic acids; and extracellular polysaccharides during feed fermentation, which improve the palatability and nutritional value of feeds and promote the digestion and absorption of nutrients, thus enhancing the growth of animals [19,43]. Canibe et al. [44] found that the ADFI and ADG of weaned piglets were improved after feeding with *Lactobacillus*-fermented liquid feed. Jonas et al. [45] found that the ADG, ADFI, and FCR of calves were significantly increased after feeding with *Enterococcus*-fermented feed. In this study, we found that the ADG of the various treatment groups did not differ significantly after feeding on different feeds; however, the ADFI was significantly lower in group L, and the FCR was significantly lower in group MIX compared with the control group. The FCR was the amount of feed consumed by the piglets to gain one kilogram of weight, which was used to evaluate the growth performance of the piglets in each group. In group MIX, representing feed fermented with potentially probiotic *L. plantarum* QP28-1a and *B. subtilis* QB8a, the FCR may have been reduced because mixed fermentation can increase the beneficial metabolites in feed, improve its nutrient level, and promote digestion. The growth-promoting effect of fermented feeds has also been associated with the strain used for fermentation, the fermentation conditions, the bacterial dose, the nutrient composition, and the growth stage of the pigs, which affected the results in different studies [46].

Probiotic-fermented feeds can improve the diversity and structure of the intestinal microbiota and inhibit the growth and reproduction of pathogens, thus contributing to the maintenance of intestinal microbiota balance and health [47]. Tajima et al. [41] found that feeding with *Lactobacilli*-fermented feeds increased the Shannon and Chao indices of the intestinal microbiota in weaned piglets, which is consistent with our findings. However, the addition of antibiotics decreased the α-diversity in group A (Figure 3). Similarly, Fouhy et al. [48] found a decrease in the diversity of the gut microbiota in neonates treated with antibiotics, suggesting the advantage of fermented feeds in preserving gut microbiota diversity. LAB produce a variety of organic acids (mainly lactic acid) during fermentation, which decrease the feed pH and inhibit the growth of certain harmful bacteria. After ingestion by animals, acid-resistant bacteria such as *Lactobacilli* and *Bifidobacteria* and other probiotics enter and colonize the intestinal tract to become the predominant bacteria, decreasing the number of harmful bacteria and thus improving the microbiota structure and the micro-ecological environment. Van Winsen et al. [49] found that feeding with *L. plantarum*-fermented feed significantly reduced the number of *Escherichia coli* and *Salmonella* in swine feces and increased the number of LAB. Our results showed that the addition of probiotics, especially *L. plantarum*, not only inhibited the growth of *Clostridium* and *Streptococcus* in the intestinal tracts of Bamei pigs, but also increased the relative abundance of *Lactobacillus* (Figure 5). Notably, *Streptococcus*, and opportunistic pathogens such as *Streptococcus mutans*, which causes dental caries [50], and Group B *Streptococcus* present in the intestinal and genitourinary tracts [51] virtually disappeared from groups L, B, and MIX by day 50 (Figure 5D). Thus, the fermented feeds effectively reduced the risk of these diseases. In summary, probiotic-fermented feeds improved the intestinal microbiota of the Bamei pigs.

Cytokines mediate and regulate immune responses [52]. Probiotic-fermented feed can not only promote the absorption of nutrients and the balance of the intestinal microbiota in animals but also improve their resistance to invasion by foreign microorganisms and immunity, especially intestinal immunity [53]. In general, probiotic-fermented feeds affect animal immunity through three routes: pro-inflammatory cytokines, anti-inflammatory cytokines, and typical immune-related signaling pathways. On the one hand, when an organism is infected by pathogens, the expression levels of pro-inflammatory cytokines such as IL-1β and TNF-α are up-regulated, producing a mucosal inflammatory response [54]. On

the other hand, probiotics can induce the differentiation of Th2 cells, eliciting the expression of anti-inflammatory cytokines such as IL-2 and IL-10 and reducing the damage caused by inflammatory responses [55]. Sanchez-Muñoz et al. [56] reported that *Lactobacillus*-fermented feeds reduced the expression of pro-inflammatory cytokines IL-1β and TNF-α in fish intestines, alleviating inflammation. De-Simone et al. [57] also found that *Bifidobacterium infantis* down-regulated IL-1β and TNF-α and up-regulated IFN-γ and IL-10 in the gut, thereby alleviating inflammation. In addition to the cytokines mentioned above, signaling pathways, especially the TLR4/MyD88/NF-κB signaling pathway, are also involved in regulating immune and inflammatory responses [58]. We found (Figure 2) that the pro-inflammatory cytokine IL-1β was significantly decreased ($p < 0.05$), the anti-inflammatory cytokines IL-2 and IL-10 significantly increased ($p < 0.05$), and the TLR4/MyD88/NF-κB signaling pathway significantly down-regulated in groups L, B, and MIX, suggesting that fermented feeds supplemented with the potentially probiotic *L. plantarum* QP28-1a or *B. subtilis* QB8a improved the immunity of Bamei pigs. Group MIX showed the most pronounced immune-enhancing effect, even stronger than that of the antibiotic group.

## 5. Conclusions

Supplementation with the potentially probiotic mixed *L. plantarum* QP28-1a and *B. subtilis* QB8a significantly reduced the pH and NDF and ADF contents of fermented feeds, thus improving their digestibility for Bamei pigs. Additionally, Coliform bacteria, *Clostridium*, and aerobic bacteria were effectively inhibited. The potentially probiotic-fermented feed not only achieved reduced feed-conversion ratios but also improved immunity by increasing the production of anti-inflammatory cytokines IL-2 and IL-10 and decreasing the production of pro-inflammatory cytokine IL-1β and typical TLR4/MyD88/NF-κB inflammation-related signaling pathways. The 16s rDNA high-throughput sequencing results showed that *L. plantarum* QP28-1a and *B. subtilis* QB8a-fermented feeds improved the diversity of intestinal microbiota, significantly reduced the relative abundance of harmful bacteria such as *Clostridium* and *Streptococcus*, and significantly increased the relative abundance of beneficial bacteria such as *Lactobacillus* and *Prevotella*, which was conducive to maintaining the intestinal health of pigs. In conclusion, supplementation with mixed *L. plantarum* QP28-1a and *B. subtilis* QB8a improved the feed quality and the immunity of Bamei pigs, and optimized the intestinal microbiota structure of Bamei pigs, demonstrating promising application prospects.

**Author Contributions:** Conceptualization, Z.T., G.W., Y.C. and H.P.; methodology, Z.T., L.W., J.C., J.H. and M.Z.; software, G.W., J.C., L.W. and J.H.; validation, G.W., Y.G., Y.C. and G.Q.; formal analysis, J.C., G.W. and Y.D.; investigation, J.C., M.Z. and L.W; resources, L.W. and Y.D.; data curation, H.P. and Y.C; writing—original draft preparation, J.C.; writing—review and editing, J.C., Z.T., L.W. and H.P.; visualization, G.W. and L.W.; supervision, G.Q., Y.C. and Z.T.; project administration, G.W., Y.C. and Z.T.; funding acquisition, Z.T. All authors have read and agreed to the published version of the manuscript.

**Funding:** This work was supported by the Qinghai Province Key R&D and Transformation Plan of China (No. 2023-NK-137).

**Institutional Review Board Statement:** The experiment was approved by the Institutional Animal Care and Use Committee at Zhengzhou University under the certificate number ZZUIRB2021-111, Zhengzhou, China (28 October 2021).

**Informed Consent Statement:** Not applicable.

**Data Availability Statement:** The datasets presented in this study can be found in online repositories. The 16S rRNA gene sequence of *Lactiplantibacillus plantarum* QP28-1a and *Bacillus subtilis* QB8a used to support the findings of this study were deposited in the GenBank repository with accession numbers OM049403 and OQ703036, respectively (http://www.ncbi.nlm.nih.gov, accessed on 12 August 2023).

**Conflicts of Interest:** The authors declare no conflict of interest.

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
