# Peer review of "Effect of Mixed Lactiplantibacillus plantarum- and Bacillus subtilis-Fermented Feed on Growth, Immunity, and Intestinal Health of Weaner Pigs"

_fermentation, doi:10.3390/fermentation9121005_

Round 1

Reviewer 1 Report

Comments and Suggestions for Authors

The paper is readable, but suffers from excessive number of minor errors in language and presentation. Input from an English Editor is needed.

I also have number of technical queries.

The use of ‘Bamei’ breed in the title and (throughout the) text is misleading. Crossbred (Bamei x Large White x Duroc- see L141) pigs were used and the term ‘Bamei’ pig cannot be used. Simply use the term ‘weaner pigs’

INTRODUCTION: Wordy and contains lot of redundant statements. For example, Para 1that focuses on Bamei pig must be deleted, for the reason provided above. Para 2 could be reduced to few sentences. Paras 3 and 4 must be shortened. The questions in Para 4 are unnecessary – simply state the aims of the study.

The study has used only 3 replications (of 5 pigs in each; L142); this does not give any statistical validity to the results generated – a serious flaw.

TABLE 1: basal feed, not basic feed – change here and elsewhere.

What was the silage? Rape stalk? Stone powder?. Metabolizable energy, not metabolic energy!; Unit – KC/kg ???. Need calculated values for M+C and Thr.

L164: why on 5 different dates?. Does this add more insight/

Section 2.5: remove the sub-headings

L200: what is ‘ketone body weight’?

L206: measured in what samples?. How collected?; when?

L214: why in fecal samples, and not in cecal ?.

Seems like, all microbial and immune measures are based on 3 replicates; sampling and stats used must be clearly stated.

Carefully check the references - many are not in correct format

Comments on the Quality of English Language

Readable

Author Response

For research article

Response to Reviewer 1 Comments

1. Summary

Thank you very much for taking the time to review this manuscript, and we have carefully revised the whole manuscript. Please find the detailed responses below and the corresponding revisions in track changes in the re-submitted files.

2. Questions for General Evaluation

Reviewer’s Evaluation

Response and Revisions

Does the introduction provide sufficient background and include all relevant references?

Yes/Can be improved/Must be improved/Not applicable

Thank you very much for your suggestions. We have carefully revised the whole article according to all the comments and suggestions you have made and have used the English editing service provided by MDPI to make detailed revisions to improve the English level of this article.

Are all the cited references relevant to the research?

Yes/Can be improved/Must be improved/Not applicable

Is the research design appropriate?

Yes/Can be improved/Must be improved/Not applicable

Are the methods adequately described?

Yes/Can be improved/Must be improved/Not applicable

Are the results clearly presented?

Yes/Can be improved/Must be improved/Not applicable

Are the conclusions supported by the results?

Yes/Can be improved/Must be improved/Not applicable

3. Point-by-point response to Comments and Suggestions for Authors

Comments 1: [The paper is readable, but suffers from excessive number of minor errors in language.]

Response 1: [Thank you for your comments and suggestions on this article. Considering the number of minor language errors in the paper, we have applied for the professional English editing service provided by MDPI and have carefully revised the article according to their revision requirements. All detailed revisions throughout the article have been made using revision mode and are marked yellow in the resubmitted manuscript. In addition, we have received a certificate of editing in English, as shown in Figure 1. Many thanks to them for professionally and carefully revising our article.]

Figure 1. English-Editing-Certificate

Comments 2: [The use of ‘Bamei’ breed in the title and (throughout the) text is misleading. Crossbred (Bamei x Large White x Duroc- see L141) pigs were used and the term ‘Bamei’ pig cannot be used. Simply use the term ‘weaner pigs’.]

Response 2: [Thank you for pointing this out. We agree with this comment. Therefore, we have removed the ‘Bamei piglets’ from the title and replaced it with the term ‘weaner pigs’. Please see page 1, L4 in the revised manuscript. We have removed the vast majority of the article about the ‘Bamei’ pig in consideration of the misleading throughout the text, please see page 1, paragraph 1 of introduction, and L42-L52.]

Comments 3: [INTRODUCTION: Wordy and contains lot of redundant statements. For example, Para 1that focuses on Bamei pig must be deleted, for the reason provided above. Para 2 could be reduced to few sentences. Paras 3 and 4 must be shortened. The questions in Para 4 are unnecessary – simply state the aims of the study.]

Response 3: [Thank you for your suggestion, we have removed the Para 1 and all the questions in Para 4. Please see page 1-2, paragraph 1, and L42-52; page 3, paragraph 1, and L117-123. We revised and abbreviated the second paragraph regarding antibiotics and the third paragraph regarding probiotics, please see page 2-3, paragraph 2-3, and L58-70, L85-98, L106-110. The aims of the study were briefly stated in Para 4, please see page 3, paragraph 2, and L124-129.]

Comments 4: [The study has used only 3 replications (of 5 pigs in each; L142); this does not give any statistical validity to the results generated – a serious flaw.]

Response 4: [Thank you for pointing this out. We apologize for the confusion caused by not indicating clearly the sampling process and the number of samples in the original article, especially the number of samples in the five treatment groups was not clearly described. In fact, we divided the 60 pigs into five groups of 12 replicates each and assigned the twelve pigs in each group to three pens. We have modified the description of the design of the 60-pig grouping, please see page 4, paragraph 3, and L164. Figure 2 presents a clearer visualization of our experimental groupings and replications.]

Figure 2. Schematic diagram of experimental grouping and pig pen distribution

Comments 5: [TABLE 1: basal feed, not basic feed – change here and elsewhere.]

Response 5: [Thank you for pointing this out. We have replaced all the basic in the text with basal. Please see page 3, paragraph 4, and L143; page 3, Table 1, and L146; page 4, Table 2, and L156.]

Comments 6: [What was the silage? Rape stalk? Stone powder?. Metabolizable energy, not metabolic energy!; Unit – KC/kg ???. Need calculated values for M+C and Thr.]

Response 6: [Thank you for your suggestions and corrections, sorry we didn't describe it clearly. The silage added to the basal diet is alfalfa silage. We have replaced the ‘silage’ with ‘alfalfa silage’, please see page 4, Table 1, and L146. We have replaced the ‘metabolic energy’ with ‘Metabolizable energy’, please see page 4, Table 1, and L146, and the unit has been changed to Kcal/kg. This study is a continuous project lasting three years, and the basal feed we used was supplied by the researchers from the Bamei Pig Original Breeding Base of Huzhu County in Qinghai province, who have tested and calculated the metabolizable energy of the basal feed, and some of the results have been published [1].]

[1] Wu, G.F.; Tang, X.J.; Fan, C.; Wang, L.; Shen, W.J.; Ren, S.E.; Zhang, L.Z.; Zhang, Y.M. Gastrointestinal Tract and Dietary Fiber Driven Alterations of Gut Microbiota and Metabolites in Durco x Bamei Crossbred Pigs. Front Nutr 2022, 8, doi:ARTN 80664610.3389/fnut.2021.806646.

Comments 7: [L164: why on 5 different dates?. Does this add more insight.]

Response 7: [Thank you for pointing this out. In this study, the feed was fermented for seven days and then fed to the Bamei pigs, which were fed for up to 100 days before slaughtering. Considering the long experimental period and the dynamic process of microbiota changes in the feed, the microbial counts were sampled and analyzed before, during and after the end of the experimental period in order to have a clearer understanding of the dynamic changes of microbiota in the fermented feed as well as the benefits of the potential probiotic additions to the feed. In fact, a total of 17 batches were sampled throughout the experimental cycle, and to avoid an overly lengthy article, we present the results of the study at only five time points.]

Comments 8: [Section 2.5: remove the sub-headings.]

Response 8: [We agree. We have removed subheading Section 2.5, and merged the contents of section 2.5 into section 2.4. Please see page 5, paragraph 2, and L199, L189.]

Comments 9: [L200: what is ‘ketone body weight’?]

Response 9: [Thank you for pointing this out. Ketone weight means the carcass weight of fattening pigs after slaughter and shaving, removing the head, hooves, tail and viscera, and retaining the plate oil and kidneys. It is an important item used to rate the level of meat production and carcass quality.]

Comments 10: [L206: measured in what samples?. How collected?; when?.]

Response 10: [Thank you for pointing this out. We have added content about sampling, collection and time of immune factors, please see page 6, paragraph 1, and L238-241 in the revised manuscript.]

Comments 11: [L214: why in fecal samples, and not in cecal ?]

Response 11: [Thank you for pointing this out. We took samples in the cecum and corrected the description in the text, see page 6, paragraph 1, and L238-241. We fed for 100 days before slaughtering and tested the production of immune factors at multiple time points and organ tissues. Cecum contents, i.e. feces, were sampled on day 50 of feeding, and spleen, liver, jejunum, and cecum samples were taken for testing after 100 days of slaughter, and the production of immune factors in cecum samples was tested in this article.]

Comments 12: [Seems like, all microbial and immune measures are based on 3 replicates; sampling and stats used must be clearly stated.]

Response 12: [Thank you for pointing this out. We have checked, supplemented and revised all descriptions of sampling and stats in the manuscript, please see page 5, paragraph 2, and L197-198; page 5, paragraph 3, and L202; page 5, paragraph 8, and L224; page 6, paragraph 1, and L240-241; page 6, paragraph 2, and L243-245.]

Comments 13: [Carefully check the references - many are not in correct format.]

Response 13: [Thanks to your reminder, we have carefully checked and corrected all references for incorrect format.]

4. Response to Comments on the Quality of English Language

Response: Considering the number of minor language errors in the paper, we have applied for the professional English editing service provided by MDPI and have carefully revised the article according to their revision requirements. All detailed revisions throughout the article have been made using revision mode and are marked yellow in the resubmitted manuscript. In addition, we have received a certificate of editing in English, as shown in Figure 1. Many thanks to them for professionally and carefully revising our article.

Reviewer 2 Report

Comments and Suggestions for Authors

The manuscript by Chen et al, shows the effect of bacteria used for fermentation of pig feed using either Lactiplantibacillus plantarum, Bacillus subtilis or a mix and compared to a basal diet and antibiotic-added feed. They measured several parameters on feed as well as pig growth and microbial diversity of pig-feces.  Several issues draw my attention. First, although the title of the manuscript does not mention that bacteria used in their study are probiotics, the rest of the paper considered them as probiotics. Authors need to be careful using the term probiotic, since bacteria used in this study cannot be considered as ones. If this reviewer is wrong, please provide proof these bacteria used in this study are properly deposited and received category at least as GRAS. I suggest use the term potentially probiotic or similar. In this regard, authors should indicate the source of the bacteria used in this study, the reference given (Pang et al, 2011) is for the isolation of Lactobacilli but not that of B. subtilis.

Another issue detected, is the use of the term “flora” thorough the text, please change to microbiota, since the term microflora is not proper for the scientific community.

One confusing part of the manuscript is the preparation of the fermented feed. In the Methods section (page 3) authors state that fermentation took place for seven (7) days; yet for number of microorganism, pH, organic acid and nutrient analysis, the authors indicate they took samples for fermentation on days l(2, 10, 20, 30, 50 and even 60). Which one is correct? Was the fermentation that long? was a refeeding of fermentation? If it was only for seven days, under what conditions was the feed kept after fermentation?

The paper is not clear on the use of both strains for fermentation, were they mixed just for comparison? or the authors wanted to show an advantage on the use of the mix? if so, please indicate that in the conclusions

A minor but not least important issue, is the use of italics for scientific names, please carefully revise the manuscript and change accordingly. I point some, but I stopped since there were too many

Other issues noted are listed below

L39 “named for its inverted Chinese character "eight" on the forehead” it is a curious fact, but it could be dispensable from the text

Line 80 Ref 15 talks about prebiotics, need to change to a reference for probiotics, perhaps Hill et al, doi:10.1038/nrgastro.2017.75

L81 Ref 16 outdated classification

L83 … and aquaculture to provide relief from diarrhea. Please separate the terms the relief from diarrhea is in animal feed not in aquaculture, please indicate the benefits for aquaculture

L86-90 authors state that “the metabolites or bacterial fluids of LAB have a significant bacteriostatic effect on both Gram-positive and negative bacteria, and produce antimicrobial substances, which mainly include organic acids, hydrogen peroxide, aloe-emodin, vinyl acetate, diacetyl, antimicrobial peptides, and bacteriocins” PLEASE REVISE, how metabolites produced from LAB can produce antimicrobial substances? Besides, the references 17-19 are related to assessment of probiotic potential, but says nothing to the type of substances produced

L90, 97 pathogene… pathogen (and many other places thorough the text)

L97 the growth of the growth… the growth

L109, L110, L115 flora.. please change to microbiota, the term flora is outdated.

L121 please indicate the origin of the strains used, are they commercially available?

L152 The whole feeding cycle was for 60 days... but later in L200 it says  pigs were fed for 100 days and then slaughtered, then? feeding fermented feed for 60 days and then 40 days of CK for all the pigs? Please clarify, it is really confusing

L155 lines repeated

L165 not italicized yeast, coliform, aerobic

L205 incorrect punctuation viscera. Area

L209 TL4, TLR2, MyD88 are not immune-related cytokines... they are immune-related receptors (TLR2, -4) or adaptor (MyD88) proteins

L211 the expression of the cytokines. Production or presence would be a better term, since ELISA measures protein content, whereas the expression of cytokines is related to mRNA (typically measured by qPCR) 

L249 was shown, change to is shown

L285 Table 3. What lg stands for? Logarithm? Please use scientific notation log. Also as stated above,   I am confused with the results in Table 3. In M&M the authors state that fermentation took place for 7 days, however, in table 3 they show results for fermentation up to day 60. Please clarify. If the fermentation was for 7 days, how was the feed kept afterwards. What happen with Table 3, are those the results for the feed under storage conditions?

L370 similar to L211

L379 could you please increase the size of the lettering used for statistical significance. Also include that information on the Figure description. As for figure legend¿, since the coloring is the same, perhaps they can be described rather than put in every single figure, it is way to small to visualize without zooming it.

L404 L. plantarum B subtilis, please italicize

L620 Reference 7, Moore in capital letter

Comments on the Quality of English Language

I strongly suggest a revised description of methods section, since it is crucial to understand the experimental settings. The same for the experimental design, feeding, slaughtering

Author Response

For research article

Response to Reviewer 2 Comments

1. Summary

Thank you very much for taking the time to review this manuscript, and we have carefully revised the whole manuscript. Please find the detailed responses below and the corresponding revisions in track changes in the re-submitted files.

2. Questions for General Evaluation

Reviewer’s Evaluation

Response and Revisions

Does the introduction provide sufficient background and include all relevant references?

Yes/Can be improved/Must be improved/Not applicable

Thank you very much for your suggestions. We have carefully revised the whole article according to all the comments and suggestions you have made and have used the English editing service provided by MDPI to make detailed revisions to improve the English level of this article.

Are all the cited references relevant to the research?

Yes/Can be improved/Must be improved/Not applicable

Is the research design appropriate?

Yes/Can be improved/Must be improved/Not applicable

Are the methods adequately described?

Yes/Can be improved/Must be improved/Not applicable

Are the results clearly presented?

Yes/Can be improved/Must be improved/Not applicable

Are the conclusions supported by the results?

Yes/Can be improved/Must be improved/Not applicable

3. Point-by-point response to Comments and Suggestions for Authors

Comments 1: [The manuscript by Chen et al, shows the effect of bacteria used for fermentation of pig feed using either Lactiplantibacillus plantarum, Bacillus subtilis or a mix and compared to a basal diet and antibiotic-added feed. They measured several parameters on feed as well as pig growth and microbial diversity of pig-feces.  Several issues draw my attention. First, although the title of the manuscript does not mention that bacteria used in their study are probiotics, the rest of the paper considered them as probiotics. Authors need to be careful using the term probiotic, since bacteria used in this study cannot be considered as ones. If this reviewer is wrong, please provide proof these bacteria used in this study are properly deposited and received category at least as GRAS. I suggest use the term potentially probiotic or similar. In this regard, authors should indicate the source of the bacteria used in this study, the reference given (Pang et al, 2011) is for the isolation of Lactobacilli but not that of B. subtilis.]

Response 1: [Thanks to your suggestion, we have replaced the descriptions of all the bacteria used in our study with the term potentially probiotic. please see page 1, abstract, and L23, L28, L30, L37; page 3, paragraph 2, and L124, L127; page 8, paragraph 1, and L332, L334; page 13, paragraph 1, and L448; page 17, paragraph 1, and L518; page 17, paragraph 2, and L522; page 18, paragraph 2, and L561; page 19, paragraph 2, and L619; page 19, conclusions, and L624, L628. We have supplemented the details of the source of the bacteria used in this study and the reference on methods of activation and preparation of B. subtilis., please see page 3, paragraph 3, and L132-142 in the revised manuscript.]

Comments 2: [Another issue detected, is the use of the term “flora” thorough the text, please change to microbiota, since the term microflora is not proper for the scientific community.]

Response 2: [Thank you for pointing this out. We have replaced all "flora" in the manuscript with the term microbiota. please see page 1, abstract, and L22, L34; Keywords, L38; page 3, paragraph 2, and L125; page 6, Results, and L280; page 13, paragraph 1, and L438, L449; page 15, paragraph 1, and L478; page 17, paragraph 1, and L520; page 18, paragraph 3, and L570, L572, L574, L578, L584; page 19, paragraph 1, and L595; page 19, paragraph 2, and L598; page 19, conclusions, and L634, L640.]

Comments 3: [One confusing part of the manuscript is the preparation of the fermented feed. In the Methods section (page 3) authors state that fermentation took place for seven (7) days; yet for number of microorganisms, pH, organic acid and nutrient analysis, the authors indicate they took samples for fermentation on days (2, 10, 20, 30, 50 and even 60). Which one is correct? Was the fermentation that long? was a refeeding of fermentation? If it was only for seven days, under what conditions was the feed kept after fermentation?]

Response 3: [Thank you for pointing this out. We apologize for not describing the whole experiment clearly. We have added a description of the experimental cycle, please see page 4, paragraph 3, and L171-172. We started to feed the weaned piglets after 7 days of fermentation and continued to feed them for 100 days until slaughter. All feeds used in the experiment were from the same batch prepared at the beginning of the experiment (3 tons of feeds were prepared for each treatment group), and all the feeds were kept in clean plastic buckets at room temperature. Samples were taken at different days to measure the microorganism, pH, organic acid and nutrient of the different treatment groups. Schematic 1 shows the whole fermentation and feeding process more clearly.]

Figure 1. Schematic diagram of entire fermentation and feeding process

Comments 4: [The paper is not clear on the use of both strains for fermentation, were they mixed just for comparison? or the authors wanted to show an advantage on the use of the mix? if so, please indicate that in the conclusions]

Response 4: [Thank you for pointing this out. The reason for setting up the mixed fermentation treatment group was, on the one hand, the combined consideration of the inhibitory advantage of LAB against pathogens and the ability of Bacillus to hydrolyze protein and cellulose to improve the nutrition of feed, which has been described in detail in the introduction. On the other hand, it was to explore whether the mixed fermentation had a synergistic effect compared to adding LAB or Bacillus alone, which could reduce the number of pathogenic bacteria through LAB and reduce the cellulose content by utilizing Bacillus to improve the quality of fermented feeds. The results also showed the best effect of mixed fermentation, mixed fermented feed significantly reduced the NDF and ADF, significantly reduced the FCR of the feed, improved the immunity and improved the intestinal microbiota. Please see page 19, conclusions, and L625-L639.]

Comments 5: [A minor but not least important issue, is the use of italics for scientific names, please carefully revise the manuscript and change accordingly. I point some, but I stopped since there were too many]

Response 5: [Thanks to your reminder, we have carefully revised the manuscript and changed all that needed to be italicized.]

Comments 6: [L39 “named for its inverted Chinese character "eight" on the forehead” it is a curious fact, but it could be dispensable from the text.]

Response 6: [Thank you for pointing this out. Considering that the use of ‘Bamei’ breed in the title and throughout the text is misleading and cause lot of redundant statements, we have removed the introduction about the Bamei pig, as requested by another reviewer. Please see page 1-2, Introduction, and L42-L52.]

Comments 7: [Line 80 Ref 15 talks about prebiotics, need to change to a reference for probiotics, perhaps Hill et al, doi:10.1038/nrgastro.2017.75]

Response 7: [Thanks to your suggestion, we have replaced the reference with Hill et al. doi:10.1038/nrgastro.2017.75. Please see page 2, paragraph 2, and L85 in the revised manuscript.]

Comments 8: [L81 Ref 16 outdated classification]

Response 8: [Thank you for pointing this out. We have removed this outdated classification and shortened the introduction by abbreviating the description about the probiotic classification, please see page 2, paragraph 2, and L84-88 in the revised manuscript.]

Comments 9: [L83 … and aquaculture to provide relief from diarrhea. Please separate the terms the relief from diarrhea is in animal feed not in aquaculture, please indicate the benefits for aquaculture]

Response 9: [Thank you for pointing this out. We have removed the content about aquaculture as it is not relevant to animal feed. Please see page 2, paragraph 2, and L89-90 in the revised manuscript.]

Comments 10: [L86-90 authors state that “the metabolites or bacterial fluids of LAB have a significant bacteriostatic effect on both Gram-positive and negative bacteria, and produce antimicrobial substances, which mainly include organic acids, hydrogen peroxide, aloe-emodin, vinyl acetate, diacetyl, antimicrobial peptides, and bacteriocins” PLEASE REVISE, how metabolites produced from LAB can produce antimicrobial substances? Besides, the references 17-19 are related to assessment of probiotic potential, but says nothing to the type of substances produced]

Response 10: [Thank you for pointing this out. We have revised our content on antimicrobial substances produced by LAB and added relevant references, please see page 2, paragraph 2, and L93-98 in the revised manuscript. Considering the wordy redundant statements of the introduction, we have deleted or abbreviated part of the introduction about the Bamei pigs, probiotics and fermented feeds as requested, and changed this paragraph to ‘LAB can also produce organic acids, bacteriocins, and other antimicrobial substances’, and the added references [1,2] describes that LAB inhibit the growth of pathogens by producing lactic acid or bacteriocins.]

[1] Magnusson, J.; Schnürer, J. subsp.: strain Si3 produces a broad-spectrum proteinaceous antifungal compound. Appl Environ Microb 2001, 67, 1-5, Doi 10.1128/Aem.67.1.1-5.2001.

[2] Missotten, J.A.M.; Michiels, J.; Degroote, J.; De Smet, S. Fermented liquid feed for pigs: an ancient technique for the future. J Anim Sci Biotechno 2015, 6, doi:Artn 410.1186/2049-1891-6-4.

Comments 11: [L90, 97 pathogene… pathogen (and many other places thorough the text)]

Response 11: [Thanks to your reminder, we have carefully revised the manuscript and changed all ‘pathogene’ to ‘pathogens’. Please see page 2-3, paragraph 2, and L75, L78 L90, L91, L98, L108; page 17, paragraph 1, and L529; page 18, paragraph 3, and L570, L590; page 19, paragraph 2, and L603 in the revised manuscript]

Comments 12: [L97 the growth of the growth… the growth]

Response 12: [Thank you for pointing this out. We have deleted the redundant 'of the growth of the animal'. Please see page 3, paragraph 1, and L105 in the revised manuscript.]

Comments 13: [L109, L110, L115 flora. please change to microbiota, the term flora is outdated]

Response 13: [Thank you for pointing this out. We have replaced all "flora" in the manuscript with the term microbiota. please see the Response 2.]

Comments 14: [L121 please indicate the origin of the strains used, are they commercially available?]

Response 14: [Thanks to your suggestion. We have supplemented the details of the origin of the bacteria used in this study, please see page 3, paragraph 3, and L132-142 in the revised manuscript.]

Comments 15: [L152 The whole feeding cycle was for 60 days... but later in L200 it says  pigs were fed for 100 days and then slaughtered, then? feeding fermented feed for 60 days and then 40 days of CK for all the pigs? Please clarify, it is really confusing]

Response 15: [Thank you for pointing this out. We apologize for not describing the whole experiment clearly. We have added a description of the experimental cycle, please see page 4, paragraph 3, and L171-172. We started to feed the weaned piglets after 7 days of fermentation and continued to feed them for 100 days until slaughter. All feeds used in the experiment were from the same batch prepared at the beginning of the experiment (3 tons of feeds were prepared for each treatment group), and all the feeds were kept in clean plastic buckets at room temperature. Samples were taken at different days to measure the microorganism, pH, organic acid and nutrient of the different treatment groups. Schematic 1 shows the whole fermentation and feeding process more clearly.]

Figure 1. Schematic diagram of entire fermentation and feeding process

Comments 16: [L155 lines repeated]

Response 16: [Thank you for the correction. We have removed the duplicate content, please see page 4, paragraph 3, and L174-176 in the revised manuscript.]

Comments 17: [L165 not italicized yeast, coliform, aerobic]

Response 17: [Thank you for the correction. We've eliminated their italics. Please see page 5, paragraph 1, and L184-185 in the revised manuscript.]

Comments 18: [L205 incorrect punctuation viscera. Area]

Response 18: [Thank you for the correction. We've corrected the punctuation and removed the redundant 'Area', please see page 5, paragraph 9, L230-231, and L225-226 in the revised manuscript.]

Comments 19: [L209 TL4, TLR2, MyD88 are not immune-related cytokines... they are immune-related receptors (TLR2, -4) or adaptor (MyD88) proteins]

Response 19: [Thank you for your suggestion. We have changed the cytokines of TLR2 and TLR4 to receptors and the cytokines of MyD88 to adaptor proteins. Please see page 6, paragraph 1, L232-233 in the revised manuscript.]

Comments 20: [L211 the expression of the cytokines. Production or presence would be a better term, since ELISA measures protein content, whereas the expression of cytokines is related to mRNA (typically measured by qPCR)]

Response 20: [Thank you for pointing this out. We've carefully scrutinized the full text and changed all the 'expression' to 'production'. Please see page 1, abstract, and L32; page 6, paragraph 1, and L238; page 11, paragraph 2, and L411, L413, L415, L417, L419; page 12, Figure 2, and L422; page 19, conclusions, and L630 in the revised manuscript.]

Comments 21: [L249 was shown, change to is shown]

Response 21: [Thank you for the correction. We have changed was to is, please see page 7, paragraph 1, and L281 in the revised manuscript.]

Comments 22: [L285 Table 3. What lg stands for? Logarithm? Please use scientific notation log. Also as stated above, I am confused with the results in Table 3. In M&M the authors state that fermentation took place for 7 days, however, in table 3 they show results for fermentation up to day 60. Please clarify. If the fermentation was for 7 days, how was the feed kept afterwards. What happen with Table 3, are those the results for the feed under storage conditions?]

Response 22: [Thank you for pointing this out. We have carefully checked the full text and changed all lg to scientific notation log, please see page 7, Table 3, and L318; page 7, paragraph 1, and L286, L288, L314 in the revised manuscript. We apologize for the confusion caused by the unclear description in Table 3. We have added a description of the experimental cycle, please see page 4, paragraph 3, and L171-172. We started to feed the weaned piglets after 7 days of fermentation and continued to feed them for 100 days until slaughter. All feeds used in the experiment were from the same batch prepared at the beginning of the experiment (3 tons of feeds were prepared for each treatment group), and all the feeds were kept in clean plastic buckets at room temperature. Samples were taken at different days to measure the microorganism, pH, organic acid and nutrient of the different treatment groups. Schematic 1 shows the whole fermentation and feeding process more clearly.]

Figure 1. Schematic diagram of entire fermentation and feeding process

Comments 23: [L370 similar to L211]

Response 23: [[Thank you for pointing this out. We've carefully scrutinized the full text and changed all the 'expression' to 'production'. Please see page 1, abstract, and L32; page 6, paragraph 1, and L238; page 11, paragraph 2, and L411, L413, L415, L417, L419; page 12, Figure 2, and L422; page 19, conclusions, and L630 in the revised manuscript.]

Comments 24 [L379 could you please increase the size of the lettering used for statistical significance. Also include that information on the Figure description. As for figure legend¿, since the coloring is the same, perhaps they can be described rather than put in every single figure, it is way to small to visualize without zooming it.]

Response 24: [Thank you for your suggestion. We have recreated Figure 2 as you suggested and replaced the figure in the manuscript. Please see page 12, Figure 2, and L421 in the revised manuscript.]

Comments 25: [L404 L. plantarum B subtilis, please italicize]

Response 25: [Thank you for pointing this out. We have italicized these words and double-checked all italics throughout the text. Please see page 13, paragraph 1, and L448 in the revised manuscript.]

Comments 26: [L620 Reference 7, Moore in capital letter]

Response 26: [Thank you for pointing this out. We have capitalized 'Moore' and carefully revised the reference format. Please See page 20, Reference 1, and L663.]

4. Response to Comments on the Quality of English Language

Response: Considering the number of minor language errors in the paper, we have applied for the professional English editing service provided by MDPI and have carefully revised the article according to their revision requirements. All detailed revisions throughout the article have been made using revision mode and are marked yellow in the resubmitted manuscript. In addition, we have received a certificate of editing in English, as shown in Figure 2. Many thanks to them for professionally and carefully revising our article.

Figure 2. English-Editing-Certificate

Round 2

Reviewer 2 Report

Comments and Suggestions for Authors

The authors revised the manuscript carefully and addressed almost all remarks extensively.

Here are some minor remarks still to address:

 L40 please add a coma … can, otherwise it reads as can not only (which changes the meaning of the sentence)

L75 in the sentence  “…also comprises probiotics such as Bacillus subtilis…” not all the species of Bacillus subtilus or coagulans are probiotics, some specific strains are probiotics, therefore,  I suggest changing to species “…also comprises species  considered as probiotics such as some strains of Bacillus subtilis…”

L483, at the end of the paragraph, seems to be an unfinished sentence: “…and the growth stage of the pigs, which affected the results in [46].”

Author Response

For research article

Response to Reviewer 2 Comments

1. Summary

Thank you very much for taking the time to review this manuscript, and we have carefully revised the whole manuscript. Please find the detailed responses below and the corresponding revisions in track changes in the re-submitted files.

2. Questions for General Evaluation

Reviewer’s Evaluation

Response and Revisions

Does the introduction provide sufficient background and include all relevant references?

Yes/Can be improved/Must be improved/Not applicable

Thank you very much for your suggestions. We have carefully revised the whole article according to all the comments and suggestions you have made and have used the English editing service provided by MDPI to make detailed revisions to improve the English level of this article.

Are all the cited references relevant to the research?

Yes/Can be improved/Must be improved/Not applicable

Is the research design appropriate?

Yes/Can be improved/Must be improved/Not applicable

Are the methods adequately described?

Yes/Can be improved/Must be improved/Not applicable

Are the results clearly presented?

Yes/Can be improved/Must be improved/Not applicable

Are the conclusions supported by the results?

Yes/Can be improved/Must be improved/Not applicable

3. Point-by-point response to Comments and Suggestions for Authors

Comments 1: [L40 please add a coma … can, otherwise it reads as can not only (which changes the meaning of the sentence)]

Response 1: Thank you for the reminder. We have added a coma to line 54 of the resubmitted revised manuscript. Please see page 2, paragraph 1, and line 54 in the revised manuscript. (line 41 in the revised PDF version)

Comments 2: [L75 in the sentence “…also comprises probiotics such as Bacillus subtilis…” not all the species of Bacillus subtilus or coagulans are probiotics, some specific strains are probiotics, therefore,  I suggest changing to species “…also comprises species  considered as probiotics such as some strains of Bacillus subtilis…”]

Response 2: Thank you very much for your suggestions. We have changed the description in the article as you suggested. Please see page 3, paragraph 1, and line 100 in the revised manuscript. (line 75-76 in the revised PDF version)

Comments 3: [L483, at the end of the paragraph, seems to be an unfinished sentence: “…and the growth stage of the pigs, which affected the results in [46].”]

Response 3: Thank you for pointing this out. We have completed this incomplete sentence. Please see page 18, paragraph 2, and line 568 and line 569  in the revised manuscript. (line 483 in the revised PDF version)